



# Non-target analysis using gas chromatography with time-of-flight mass spectrometry: application to time series of fourth generation synthetic halocarbons at Taunus Observatory (Germany).

Fides Lefrancois[1], Markus Jesswein[1], Markus Thoma[1], Andreas Engel[1], Kieran Stanley[1], and Tanja Schuck[1]

[1]Institute for Atmospheric and Environmental Sciences, Goethe University Frankfurt, Germany

**Correspondence:** Fides Lefrancois (lefrancois@iau.uni-frankfurt.de)

**Abstract.** Production and use of many synthetic halogenated trace gases are regulated internationally because of their contribution to stratospheric ozone depletion or to climate change. In many applications they have been replaced by shorter-lived compounds which have become measurable in the atmosphere as emissions increased. Non-target monitoring of trace gases rather than targeted measurements of well-known substances is needed to keep up with such changes in the atmospheric com-

position. We regularly deploy gas chromatography (GC) coupled to time-of-flight mass spectrometry (TOF-MS) for analysis of flask air samples and in situ measurements at the Taunus Observatory, a site in central Germany. TOF-MS record data over a continuous mass range enable a retrospective analysis of the data set, which can thus be considered a type of digital air archive. This archive can be made use of if new substances come into use and their mass spectrometric fingerprint is identified. However, quantifying new replacement halocarbons can be challenging, as mole fractions are generally low, requiring high

measurement precision and low detection limits. In addition, calibration can be demanding, as calibration gases may not contain sufficiently high amounts of newly used substances or the amounts in the calibration gas have not been quantified. This paper presents an indirect data evaluation approach for TOF-MS data, where the calibration is linked to another compound which could be quantified in the calibration gas. We also present an approach to evaluate the quality of the indirect calibration method and to select periods of stable instrument performance and well suited reference compounds. The method is applied to

three short-lived synthetic halocarbons: HFO-1234-yf, HFO-1234ze(E), and HCFO-1233zd(E). They represent replacements for longer-lived HFCs and exhibit increasing mole fractions in the atmosphere.

The indirectly calibrated results are compared to directly calibrated measurements using data from TOF-MS canister sample analysis and TOF-MS in situ measurements, which are available for some periods of our data set. The application of the indirect calibration method on several test cases can result into accuracies around 13 % to 20 %. For H(C)FOs accuracies up to 25 % are

achieved. The indirectly calculated mole fractions of the investigated H(C)FOs at Taunus Observatory range between measured mole fractions at urban Dübendorf and Jungfraujoch stations in Switzerland.



# 1 Introduction

Halocarbon measurements of atmospheric air are commonly performed with coupled gas chromatography/mass spectrometry (GC-MS). Often, quadrupole mass spectrometers are used, as they are reliable instruments with good long-term stability and linearity over a large measurement range. In Europe, the focus of regular halocarbon measurements has been mainly on clean air sites, which are part or affiliated to the Advanced Global Atmospheric Gases Experiment (AGAGE) (Prinn et al., 2018), at Jungfraujoch (3589 m ASL, Switzerland), Monte Cimone (2165 m ASL, Italy), Zeppelin Observatory (490 m ASL, Norway), and Mace Head (25 m ASL, Ireland). Combining observations from these sites with atmospheric transport models, it is possible to infer emission sources using inverse Bayesian models, as shown in several previous studies (e. g. (Keller et al., 2012; Maione et al., 2014; O'Doherty et al., 2014; Brunner et al., 2017). Central Europe, from where large emissions are estimated, is not covered well by these sites (Henne et al., 2010). Therefore, sample collection using flasks was established in 2013 at Taunus Observatory (TOB) in Germany (Hoker et al., 2015; Schuck et al., 2018). Data from Taunus Observatory are expected to improve the sensitivity of model-based emission estimates. In May 2018, the measurements were extended by installation of two-hourly in situ measurements. Both measurements series employ time-of-flight mass spectrometry (Hoker et al., 2015; Obersteiner et al., 2016b). In addition to TOF-MS, which is recording a continuous mass spectrum over the complete chromatogram, flask-air samples are quantified using quadrupole mass sepctrometry, where predefined masses are scanned at selected time intervals. Thus, known species can be evaluated, but also non-target analysis of species not in the focus at the time of measurement becomes possible. Time-of-flight data therefore represent a kind of digital air archive of atmospheric trace gas measurements.

Retrospective data analysis can be challenging, in particular if substances were not contained in calibration standards used at the time of a measurement. Here, we present an indirect calibration approach for retrospective non-target analysis of halocarbons. To verify the applicability of the indirect calibration method, it is applied to several substances analysed both in situ and in whole air flask samples. As examples of tracers for which a retrospective analysis is highly valuable, we present measurements of three short-lived hydro(chloro-)fluoroolefines H(C)FO: HFO-1234yf (2,3,3,3-tetrafluoroprop-1-ene, $CFC_3CF=CH_2$, HFC-1234yf, CAS 754-12-1), HFO-1234ze(E) (E-1,3,3,3-tetrafluoro-1-ene, $trans-CF_3CH=CHF$, HFC-1234ze(E), CAS 29118-24-9), and HCFO-1233zd(E) (E-1-chloro-3,3,3-trifluoroprop-1-ene, $trans-CF_3CH=CHCl$, HCFC-1233zd(E), CAS 102687-65-0). These hydro(chloro-)fluoroolefines are the so-called fourth generation of synthetic halocarbons. Due to their carbon double bond, HFO and HCFO are very short-lived with global average lifetimes from 10 to 46 days (Burkholder et al., 2018). This results in a very low global warming potential (GWP). In addition, the HCFO only carry very little chlorine into the stratosphere, resulting in very low ozone depletion potentials (ODP). Both saturated hydrofluorocarbons (HFC) and unsaturated HFO have no ODP (Patten and Wuebbles, 2010; Orkin et al., 2014). However, some HFO, as some HFC and HCFC, can form the very persistent and toxic trifluoroacetic acid (TFA) as the main breakdown product in the atmosphere (Burkholder et al., 2015). TFA is known to cause negative environmental impacts accumulating in water and soil (Ellis et al., 2001; Russell et al., 2012; Solomon et al., 2016).





These substances were observed in the atmosphere for the first time around 2010–2014 at Jungfraujoch and Dübendorf in Switzerland (Vollmer et al., 2015). The percentage of detectable mole fractions, the yearly mean mole fractions and the magnitude have increased at both sites after 2010 with the high mountain site Jungfraujoch generally experiencing lower mole fractions. Vollmer et al. (2015) identified the Benelux region and the western parts of Germany as source regions, therefore measurable mole fractions are expected to occur at TOB and measurements at the site are expected to have the potential to improve estimates of European emissions of H(C)FO.

In this paper we present a method which allows the quantification of absolute mole fractions of compounds which were not detectable in the calibration gas used at the time of the measurements. The available measurements from TOB are described in section 2. In section 3 we present and evaluate a new method which allows for an indirect calibration of such compounds for a retrospective quantification. This methods is then applied to H(C)FO in section 4.

## 2 Measurements

### 2.1 Site characterisation

The Taunus Observatory (TOB) is located at 50.22° N, 8.44° E and at an altitude 825 m.a.s.l. on the mountain top of the *Kleiner Feldberg*, the second highest mountain in the Hessian *Taunus* mountain range. It is located approximately 20 km northwest of Frankfurt am Main and it is situated near the Rhein-Main area. The surrounding area is characterised by a dense population and several locations of industry, including chemical industries to the south and south-west. To the north and west, the site is surrounded mainly by forest and agricultural areas. The site is often exposed to European background air approaching at higher altitudes mainly from northern directions, but also to local and regional pollution events (Schuck et al., 2018). Trace gas mole fractions therefore exhibit a high variability with somewhat higher baseline mole fractions compared to clean air sites.

### 2.2 Weekly whole air sampling

Whole air canister sampling was started at Taunus Observatory in October 2013. Details of the sample collection and the analytical procedure were described in detail by Schuck et al. (2018) and Hoker et al. (2015) and are only briefly summarised here. Air is collected pairwise approximately weekly using stainless steel flasks. Samples are analyzed in the laboratory at Goethe University Frankfurt using a gas chromatograph (Agilent 7890A) coupled to a quadrupole mass spectrometer (Agilent 5975C) and a time-of-flight mass spectrometer (TOF-MS; Markes Bench TOF-dx E-24). For quality assurance each sample is measured twice, and each double measurement is bracketed by a measurement of a calibration standard. Due to the low mole fractions of the investigated substances (range of parts per trillion (ppt)), a preceding cryofocussing enrichment unit is used (Obersteiner et al., 2016b). The sample loop, a 1/16" stainless steel tube of 10 cm length, is filled with an adsorbent material (HayeSep D, Vici Valco Inc., Meshsize 60/80) and is mounted in an aluminium block, cooled by a Stirling cooler (Global Cooling, M150) to -80 °C. During enrichment, sample flow is controlled by a mass flow controller (Bronkhorst) and sample volume is monitored by a pressure measurement inside a 4 L reference volume. After enrichment of 1 L of sample air at a





flow rate of 150 mL/min, the sample loop is heated to approx. 200 °C for 4 minutes and the enriched species are desorbed and transferred to the GC column using purified Helium (quality 6.0, purification System: Vici Valco HP2). Samples are dried prior to enrichment using a magnesium perchlorate $(Mg(ClO_4)_2)$ trap kept at 80 °C. Behind the GC column, the flow is split into the two mass spectrometers with the TOF-MS receiving approximately 40 % of the flow. In the following, only data from the

TOF-MS are used. From October 2013 to October 2018 the calibration gas used was a whole air standard filled in 2007 at Jungfraujoch and in the following named GUF-10. In November 2018 it was changed for a newer standard filled at Taunus Observatory in April 2018, named GUF-16. Mole fractions of both working standards were calibrated against an AGAGE gas standard and therefore are reported on scales of Scripps Institution of Oceanography (SIO), Swiss Federal Institute of Metrology (METAS), and Swiss Federal Laboratories for Materials Science and Technology (EMPA) (Table 1).

**2.3    Continuous in situ measurements**

Continuous in situ measurements with ambient air sampling every two hours were started at TOB in May 2018. The air intake is located 12 m above the ground and uses a downstream diaphragm pump to continuously pull air from the inlet into the laboratory. The 3/8 " inlet line is heated to 70 °C to avoid condensation and freezing. To prevent the intrusion of particles, a mud dauber (Swagelok SS-MD-4) is installed at the open end of the inlet line. Halogenated trace gases are analysed using a

gas chromatograph (Agilent 7890B), a TOF-MS (model EI-003, Tofwerk AG, Switzerland), and a preceding enrichment unit, which is similar to the enrichment unit used in the laboratory. Details are described by Obersteiner et al. (2016a) and are only briefly reviewed here. For each measurement approximately 500 mL of air are enriched in the sample loop at a sample flow of 80 mL/min. To determine the exact volume of enriched air, a mass flow controller (MFC; EL-FLOW F-201CM, Bronkhorst) and a pressure sensor (Baratron 626, 0-1000 mbar, accuracy including non-linearity 0.25 % of reading, MKS Instruments,

Germany) are used. The sample loop is flash heated to about 220 °C for 120 seconds during sample desorption. Purified helium is used as carrier gas (quality 6.0, purification System: Vici Valco HP2). Samples are dried using a $Mg(ClO_4)_2$ trap to remove water vapour. Sample inlets are mounted inside a heated box (80 °C) and are connected via 1/8 " quick connectors (Swagelok). The dryer, a 10-position-selector (model EUTA-2SD10MWE, Vici Valco Inc., USA), and a 4-port 2-position valve (model D4UWE, Vici Valco Inc., USA) are mounted inside the heated box and are connected via 1/8" stainless steel tubing. Directly

before each measurement, the dryer and tubing of the system are purged and conditioned for 1 min at a flow of 100 mL/min using the subsequent sample (air, calibration gas, etc.), bypassing the sample loop.

    Measurements of ambient air are bracketed by calibration gas measurements, giving a fully calibrated air sample every 2 hours. Following every 13th air measurement, a target standard is measured instead to monitor long-term stability of the set-up. From May 2018 to March 2019 the calibration gas used was a whole air standard filled in February 2015 at TOB (GUF-14). In

March 2019 it was changed for a newer standard also filled at TOB in April 2018 (GUF-17). mole fractions of both working standards were calibrated against an AGAGE gas standard. Mole fractions are reported on Scripps Institution of Oceanography (SIO) and Empa scales (Table 1).





## 2.4 Data evaluation

For both measurement set-ups, the integration of the chromatographic peaks is performed in a similar way as described in

Schuck et al. (2018). The signal areas $A$ of each substance are divided by the enriched sample Volume $V$ to yield a response $R$. A relative response $rR$ of each analysed substance is calculated by dividing the response of a substance in an air sample measurement ($R_{air}$) by the linearly interpolated response of the bracketing calibration gas measurements ($R_{cal}$):

$$rR = \frac{R_{air}}{R_{cal}} = \frac{A_{air}/V_{air}}{A_{cal}/V_{cal}}. \tag{1}$$

Thereafter, $rR$ is used to determine the mole fractions of the analysed substances if these are known in the calibration gas. In

case of a linear detector response, the mole fraction in an air sample, $\chi_{air}$, is determined by multiplying the relative response with the mole fraction of the calibration gas, $\chi_{cal}$:

$$\chi_{air} = \chi_{cal} \cdot rR. \tag{2}$$

For the measurements of the weekly whole air sampling programme, an automated procedure is used to filter the data based on the double analysis of samples and parallel sampling into two canisters to ensure a high-quality dataset, as described in

Schuck et al. (2018). For the in situ measurements, only one measurement and one preceding and subsequent calibration gas measurement are available. The standard gas measurements are used to determine the measurement precision by comparing each standard with the bracketing standard measurements. An average weekly precision value for each substance is derived from this. If a calibration gas measurement differs more than the average weekly $1\sigma$-precision range from the previous or subsequent calibration gas measurements, the air measurements between those differing calibration measurements will be

neglected.

## 3 Indirect calibration

### 3.1 Method concept

The need of an indirect calibration approach for short-lived H(C)FOs arises from the fact that these compounds were measurable with the TOF-MS already before calibration standards were used that contained measurable amounts of these substances.

As such, when these compounds were detectable in ambient air, the peak areas cannot be converted to mole fractions using Eq. 2 because neither numbers for $A_{cal}$ nor $rR$ are available. Therefore, a mathematical relation between a compound which is measurable in the standard and the target compounds (i. e. the H(C)FOs) is needed. Ideally, the sensitivity of the analytical system for the two different species behaves similarly, that means that the ratio of signal per amount of analyte for the two compounds is constant with time. If this is the case, the ratio of responses $R$ of two species is close to constant. In case of equal

amounts of sample ($V_{cal} = V_{air}$), the ratio can also be computed from the ratio of the signal areas ($A$). If the responses and areas



are further normalised to the mole fractions of the two species, this ratio should be the same for any sample. We refer to this ratio as relative response factor $rRF$:

$$rRF = \frac{R_2/\chi_2}{R_1/\chi_1} = \frac{A_2/\chi_2}{A_1/\chi_1}. \tag{3}$$

This relation applies to both ambient air measurements and calibration gas measurements. Eq. (3) can be rearranged to yield:

$$\chi_2 = \frac{A_2}{A_1} \cdot \frac{\chi_1}{rRF}. \tag{4}$$

Combining Eq. 4 with Eq. 2 and Eq. 1 for ambient air measurements, the mole fraction of species 2 can then be derived by

$$\chi_{\text{air},2} = \frac{A_{\text{air},2}}{A_{\text{air},1}} \cdot \frac{A_{\text{air},1}}{A_{\text{cal},1}} \cdot \frac{\chi_{\text{cal},1}}{rRF} = \frac{A_{\text{air},2}}{A_{\text{cal},1}} \cdot \frac{\chi_{\text{cal},1}}{rRF}. \tag{5}$$

Using Eq. 5, only measurements of ambient air are evaluated for species 2 and therefore that compound does not have to be present in the calibration gas in detectable amounts. The $rRF$ can be evaluated independently, but it must be stable in time.

For a full retrospective analysis of archived data, the assumption of temporal stability or $rRF$ needs to be validated first. This can be achieved by evaluating the ratios of peak areas for species present in a sample with constant mole fractions, which is measured repeatedly in time. Thus it can be evaluated based on the peak areas in the calibration gas used for the measurement. If the $rRF$ between different species is stable in time for a given measurement system, it is possible to apply the indirect calibration method. The $rRF$ for the species of interest which is not present in the standard relative to a compound which is detectable in the standard can then be derived from measurements of another sample which has detectable amounts and known

mole fractions of both species using Eq. (3).

## 3.2 Method evaluation

### 3.2.1 Relative response factor

The methodology outlined in 3.1 is based on the assumption of a constant $rRF$ in Eq. 4. In other words, the sensitivity of

the GC-MS system towards different compounds should vary in the same way and the relative sensitivity should show no drift in time. In reality, this will not be the case, as many factors influence the sensitivity. In particular after tuning the mass spectrometer or modifications of the analytical system such as replacement of filaments, columns or sample loops, changes in relative sensitivity and thus in the $rRF$ are to be expected. Thus, to evaluate the approach described above, the temporal stability of the $rRF$ needs to be investigated and periods with stable/unstable $rRF$ need to separated. In the following we will

refer to the compound which is detectable in the standard as the main reference substance. We further define an evaluation substance, which is also present in the standard and which is used to identify periods of stable $rRF$. In order to investigate how large temporal changes of the $rRF$ are and to determine periods of low variability of the $rRF$, we have investigated the





**Table 1.** System precision ($1\sigma$) of the investigated substances of the TOF-MS used for the weekly whole air sampling (prc (TOF_Lab)) and of the TOF-MS used for the in situ measurements (prc (TOF_in situ)) and their calibration scales.

| Compound | Scale | prc (TOF_Lab) | prc (TOF_in situ) |
|----------|-------|---------------|-------------------|
| HFC-32 | SIO-07 | 8.2 % | 2 % |
| HFC-125 | SIO-14 | 1.4 % | 0.9 % |
| HFC-143a | SIO-07 | 0.9 % | 1.7 % |
| PFC-318 | SIO-14 | 0.7 % | 3.3 % |
| HFC-152a | SIO-05 | 0.9 % | 1 % |
| HFC-227ea | SIO-14 | 7.1 % | - |
| HCFC-142b | SIO-05 | 0.3 % | 0.5 % |
| HCFC-133a | EMPA-13 | 2.8 % | 3.2 % |
| HFC-245fa | SIO-14 | 1.6 % | 4.3 % |
| HCFC-141b | SIO-05 | 0.8 % | 0.5 % |
| CFC-113 | SIO-05 | 4.4 % | 0.4 % |
| HFO-1234yf | METAS-17[a] | 18.2 % | 14 % |
| HFO-1234ze(E) | EMPA-13 | 6.9 % | 15.6 % |
| HCFO-1233zd(E) | EMPA-13 | 7.9 % | 14.1 % |

[a](Guillevic et al., 2018)

temporal change of $rRF$ for the combination of selected compounds listed in Table 1. Substances in Table 1 were chosen such that they have similar retention times and peak areas as the short-lived H(C)FOs of interest. In addition, we have excluded

species which are known to elute close to water vapour and thus could be affected by the humidity of the sample and the effectiveness of the sample drier, which is expected to lead to enhanced variability in sensitivity. This was, for example, the case for HCFC-141b (1,1-Dichloro-1-fluoroethane, $CH_3CCl_2F$, CAS 1717-00-6) and CFC-113 (1,1,2-Trichloro-1,2,2-trifluoroethane, $CCl_2FCClF_2$, CAS 76-13-1) in the case of the laboratory system.

Fig. 1 shows a schematic example for the identification of periods of stable $rRF$. Panel (a) shows the $rRF_{evalu}$ for two

arbitrary substances, a so called main reference and an evaluation substance, which are both detectable in the used calibration standard. To identify periods of stable $rRF_{evalu}$, for each individual measurement the number of independent measurements with an $rRF_{evalu}$ that differs by not more than 10 % is counted. The measurement with the highest number of matching data points is used as a reference and all measurements that fall outside the 10 % interval are excluded (shown as grey data points in panel (b)). If more than one measurement has the same number of matching data points, the case with the lowest standard

deviation is selected. In panels (c) and (d) the evaluation substance is replaced by a third substance, hereafter named test substance, and the $rRF_{test}$ is plotted. In panel (d) the data point selection determined above is applied to the $rRF$ between test and main substance. For comparison, panel (c) shows the selection that would be obtained if the above procedure was directly applied to the main-test-pair of substances. For three outliers with a high peak area ratio and several outliers with



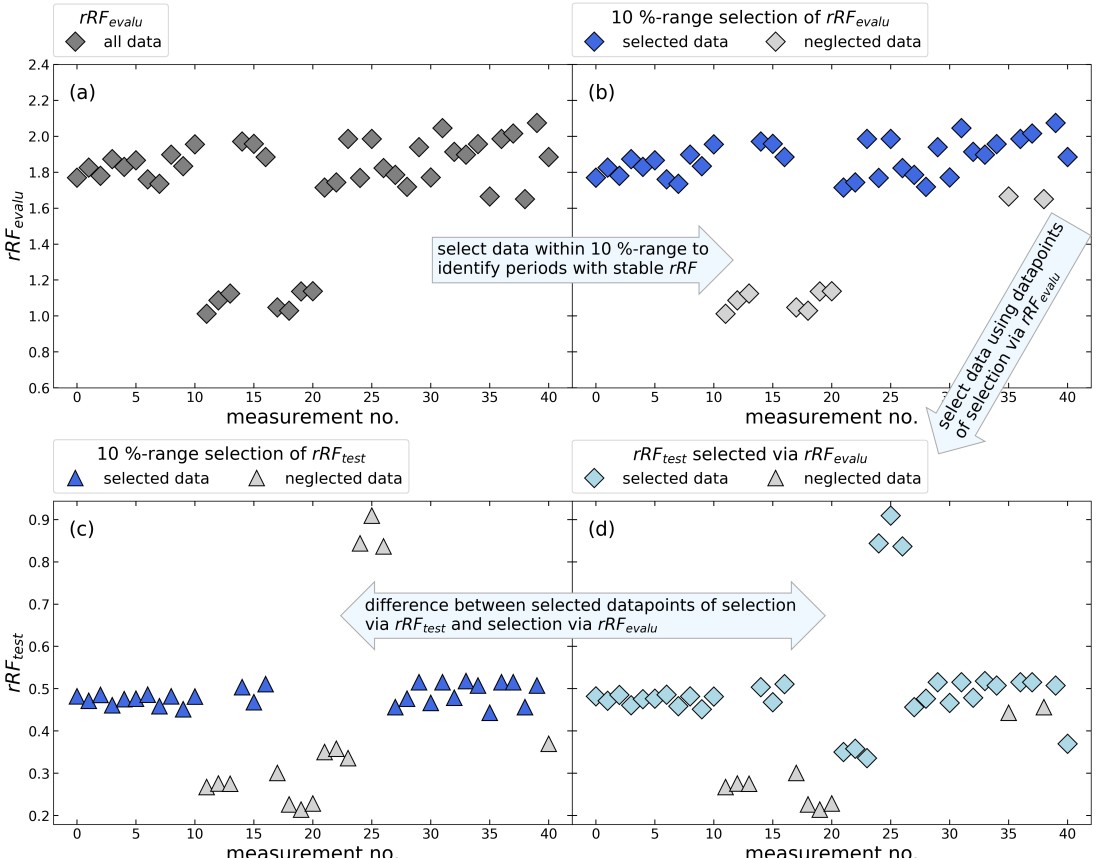

**Figure 1.** Schematic example of the identification of periods with constant $rRF$ for an undetectable substance in the calibration standard. Panels (a) and (b) show the calculated $rRF_{evalu}$ of a known main reference and a known evaluation substance. Panel (b) shows which measurements will be selected, excluding measurements where the $rRF$ differs more than $10\%$. The resulting selection of measurements should represent the periods of stable $rRF_{test}$ in panel (c) and (d), where the $rRF$ is determined using the main reference substance and an arbitrary test substance. The aim is to find a main reference and an evaluation substance, which have many measurements with a constant $rRF$ and which will represent the selection of test substances as well as possible.

low ratios a mismatch is evident. To choose the best combination of one main reference and one evaluation substance all possible combinations from the selected substances in Table 1 are investigated and tested for how well they represent known test substances.

### 3.2.2 Evaluation based on weekly sample measurements

To evaluate the stability of the $rRF$ of the laboratory GC-MS set-up used to analyse the weekly canister samples, we determined for each pair of substances from the compound selection listed in Table 1 the coefficient of determination ($r^2$) and the





195 mean absolute percentage error (MAPE). MAPE is calculated using the orthogonal distance regression fit forced through the origin and the deviation of the observed peak areas to those determined from the fit. The $r^2$ and the MAPE are both calculated for all calibration gas measurements during the measurement routines of the air samples. Fig. 2 shows an exemplary selection of correlations of peak areas for HFC-143a (a), HFC-125 (b), and HFC-227ea (c), versus HFC-152a, PFC-318, and HCFC-133a. Except HFC-227ea (column (c)), the presented substances and their comparisons of peak areas show a good correlation

200 with $r^2 > 0.95$ and MAPE < 20 %. To test which substance has in comparison to the other substances the best correlation, for each substance and its combination with all other substances the means of $r^2$ and means of MAPE are shown in Fig. 3. Except for HFC-227ea, which shows a mean $r^2$ of 0.31 and a mean MAPE of 27 %, the means of $r^2$ vary between 0.8 and 0.9 and average MAPE is below 25 % in all cases.

 As the $rRF$ is referenced relative to the mole fraction of the measured gas, this value should be independent of the mole

205 fractions and thus should also remain constant after a change of standard. Such a change of working standard occurred during the measurement time series discussed here in late 2019. Fig. 4 shows this change of standard as a dashed vertical line. While for most combinations, $rRF$ determined for the different standards do not differ significantly, a large discrepancy is found in all cases here HFC-152a is involved. The reason for this change is not known. For other main reference substances, like HFC-125 and HFC-227ea the average relative deviation of $rRF$ is below 8 %, when HFC-152a is excluded as evaluation substance.

210 Additionally, the number of measurements after selecting the periods of constant $rRF$ should remain as high as possible. Using HFC-227ea and HFC-245fa, more than half of the calibration measurements are excluded, as shown in Fig. 5. As this leads to a significant decrease in the number of air measurements for which an indirect calibration value can be derived, these substances are also less suitable as reference substances.

 The next step is the identification of periods with a constant $rRF$. Fig. 5 shows the resulting selection of suited measurement

215 periods for HFC-143a (left column) and HFC-245fa (right column) as main reference substances with HFC-125 as evaluation substance and PFC-318 as test substance. Panels (a) and (e) show the calculated $rRF$ from October 2013 to October 2018. Panels (b) and (f) in Fig. 5 show the resulting data selection for PFC-318 (left, with HFC-143a as main reference, right, with HFC-245fa as main reference). Shown is the $rRF$ of main reference and test substance. Data points which are excluded based on the evaluation substance are represented by red symbols. For comparison, panels (c) and (g) show the selection of data

220 points if the above variability filter would be applied directly to the combination of main reference and test substance. To quantify the differences between the selection of data of main reference and test substance via main reference substance and an evaluation substance we compared the relative standard deviations of the resulting filtered data sets. This is shown for all substance combinations in panels (d) and (h) (coloured points). A small range of standard deviations for a substance indicates more stable data selection and roughly correlates with a high percentage of selected data points as for example for HFC-143a.

225 Panels (d) and (h) also show the percentage of selected data points for the evaluation substances (blue vertical bars). As pointed out before, HFC-143a showed a high correlation coefficient $r^2$ and low MAPE in comparison to other substances, making it a promising candidate for the main reference substance. Using HFC-143a as main reference substance, on average 49.8 % of data points are used. In the case of HFC-245fa this decreases to only 34.6 % of data points on average.





**Figure 2.** Correlation of peak areas of exemplary substances from calibration gas measurements of phase where calibration cylinder GUF-10 was used, their coefficient of determnination ($r^2$) and the mean absolute percentage error (MAPE). Shown are the substances HFC-143a in column (a), HFC-125 in column (b), and HFC-227ea in column (c) and their comparison to HFC-152a (first row), PFC-318 (second row), and HCFC-133a (third row).

In summary, for a good indirect calibration, the main reference and an evaluation substance should show a stable $rRF$ for
a large number of measurements and also $rRF$ should be stable with a change of calibration gas. Finally, the $rRF$ of data





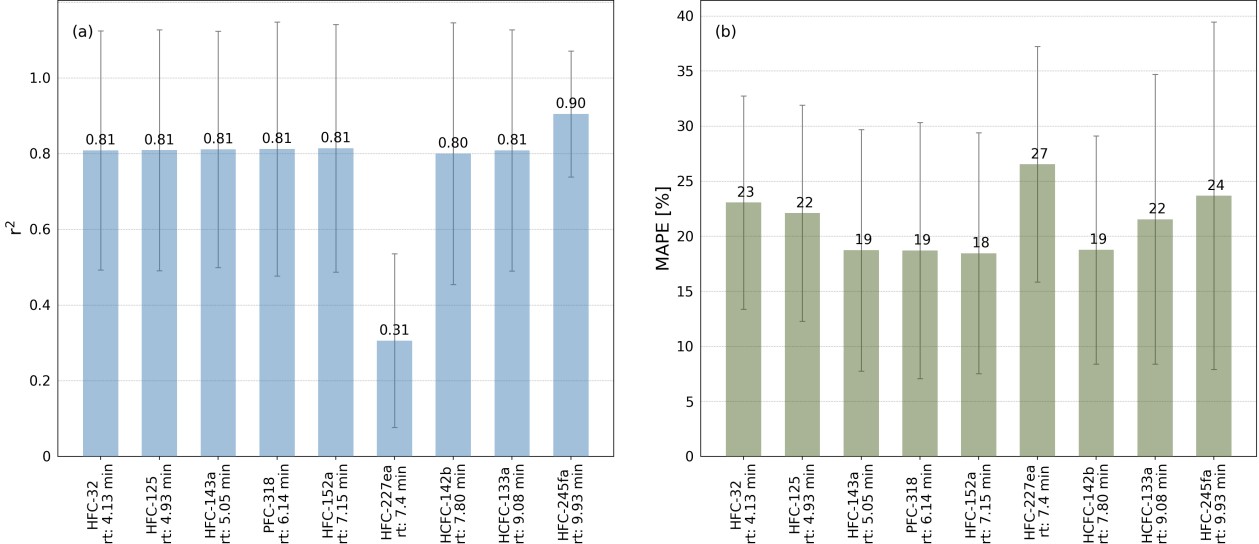

**Figure 3.** Means of the coefficient of determination ($r^2$) in panel (a) and means of mean absolute percentage error (MAPE) in panel (b) of the comparisons of the peak areas of HFC-32, HFC-125, HFC-143a, PFC-318, HFC-152a, HFC-227ea, HCFC-142b, HCFC-133a, and HFC-245fa to all other substances based on measurements of calibration standards using the laboratory TOF-MS set-up in the phase where calibration cylinder GUF-10 was used. Error bars indicate the standard deviations (1 $\sigma$).

points selected via main reference and evaluation substance should not vary too much from the $rRF$ of data points selected via main reference and test substance. Based on these criteria, we chose HFC-143a as main reference substance and HFC-125 as evaluation substance. Signal areas of HFC-143a have a high mean $r^2$ above 0.8 for all tested substances and one of the smallest mean values of MAPE with 19 %. After the application of the $\pm$ 10 % data selection criterion with HFC-125 as

evaluation substance, HFC-143a has more than 50 % of the selected data for six out of the eight tested evaluation substances. Its retention time of 7.15 min is close to that of the three target species HFO-1234yf (6.0 min), HFO-1234-ze(E) (6.8 min), and HCFO-1233zd(E) (9.6 min). Using HFC-125 as evaluation substance with HFC-143a, the difference standard deviations of the mean $rRF$ selected via the test substances and selected via itself ranges between 1 and 10 %. HFC-125 also has a large mean $r^2$ in comparison to other substances in the calibration gas measurements, and the fifth lowest mean MAPE (22 %) (cf. Fig. 3).

The next step of the method evaluation is the application to several test substances for which results of the indirect calibration can be compared to directly calibrated measurement results. As test cases to apply the indirect calibration method to we chose HFC-32, HFC-227ea, and HFC-245fa. Results are presented in Fig. 6. Shown are time series of directly and indirectly determined mole fractions (left plots) and their correlations (right plots). In this test case, mole fractions of HFC-227ea shows the best correlation with $r^2 > 0.9$ and a MAPE of 11 % (Fig. 6 (c) and (d)), whereas for the mole fractions of HFC-32 and

HFC-245fa poorer results with $r^2 = 0.79$ and $r^2 = 0.63$, respectively, are obtained. The relative average differences of the direct and indirect calculated mole fractions are given in Table 2. The $rRF$ of HFC-245fa as evaluation substance and HFC-





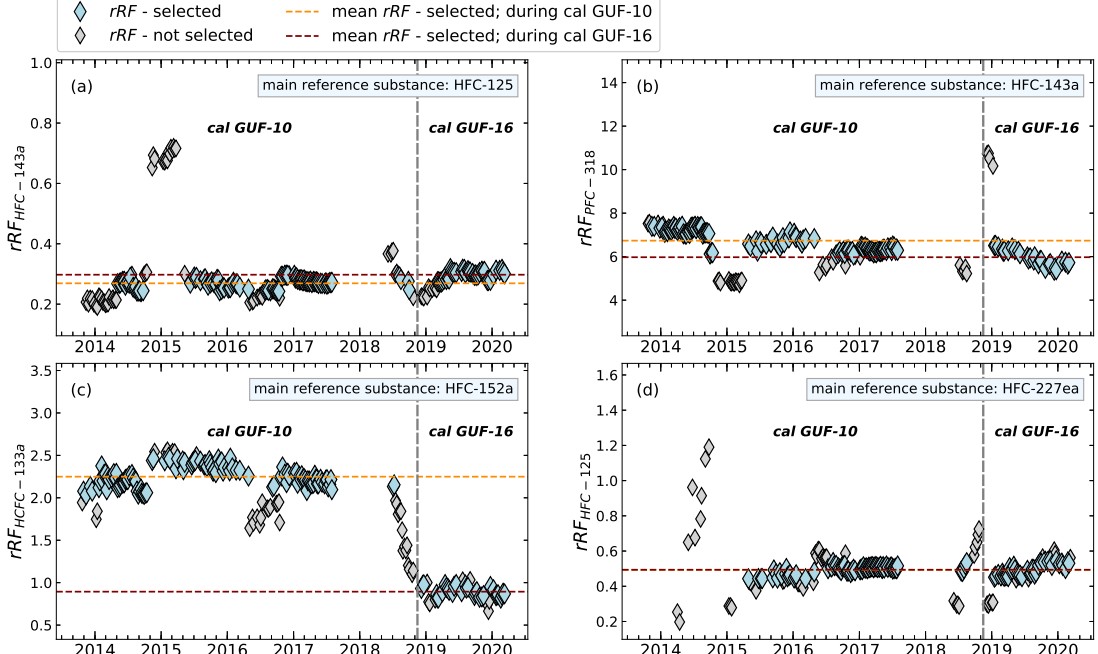

**Figure 4.** Differences of calculated and filtered $rRF$ of whole air flask sample measurements during the use of the GUF-10 and GUF-16 calibration standards. As exemplary compounds different main reference and evaluation substances are shown. The vertical grey dashed line indicates the change of the used cylinders.

143a as main reference substance has also less than 50 % of selected data within the 10 %-filter (cf. Fig. 5), which means that the calculation is applied to a large portion of data for which the criterion of a constant $rRF$ was not met. This underlines how crucial the assumption of constant instrumental sensitivity is for the indirect calibration method.

**Table 2.** Average relative differences of directly and indirectly calculated mole fractions for the whole air flask sample GC-MS measurements for the period from October 2013 to December 2018 (cf. Fig. 6). As main reference HFC-143a, as evaluation reference HFC-125 is used.

| compound | av. rel. difference | standard deviation |
|---|---|---|
| HFC-32 | 21.3 % | 20.9 % |
| HFC-227ea | 19.3 % | 13.9 % |
| HFC-245fa | 13.2 % | 9 % |



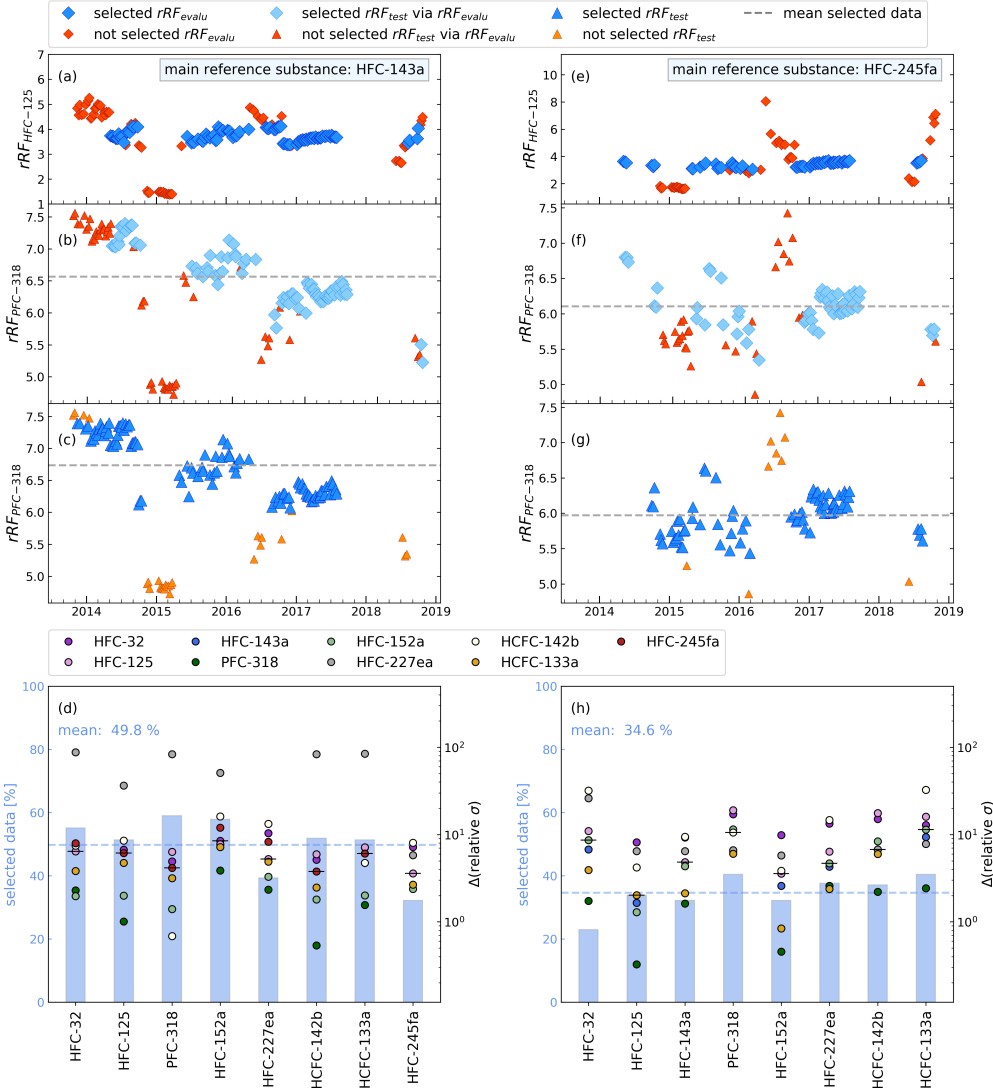

**Figure 5.** Illustration of data selection of data of the weekly flask sampling measurements using two different main reference substance (HFC-143a (a-d), HFC-245fa (e-h)) and an evaluation substance (HFC-125). Panels (a) and (e) show results of the application of the 10 %-filter. Red data points are excluded as outliers. Panels (b) and (f) illustrate which data of main reference and test substances (here PFC-318) would be chosen applying the resulting selection of panels (a) and (e). Panels (c) and (g) show the result of data selection based on the main reference and the test substance directly. The dashed lines indicate the means of the $rRF$ of the selected data. In panels (d) and (h), vertical bars show percentages of data selected for the respective main reference substance and coloured symbols show the differences of the relative standard deviations of the $rRF$ of main reference and test substance selected via main reference (HFC-143a in panel (d) and HFC-245fa in panel (h)) and all evaluation substances (x-axis) between the originally selected $rRF$ via main reference and test substance.

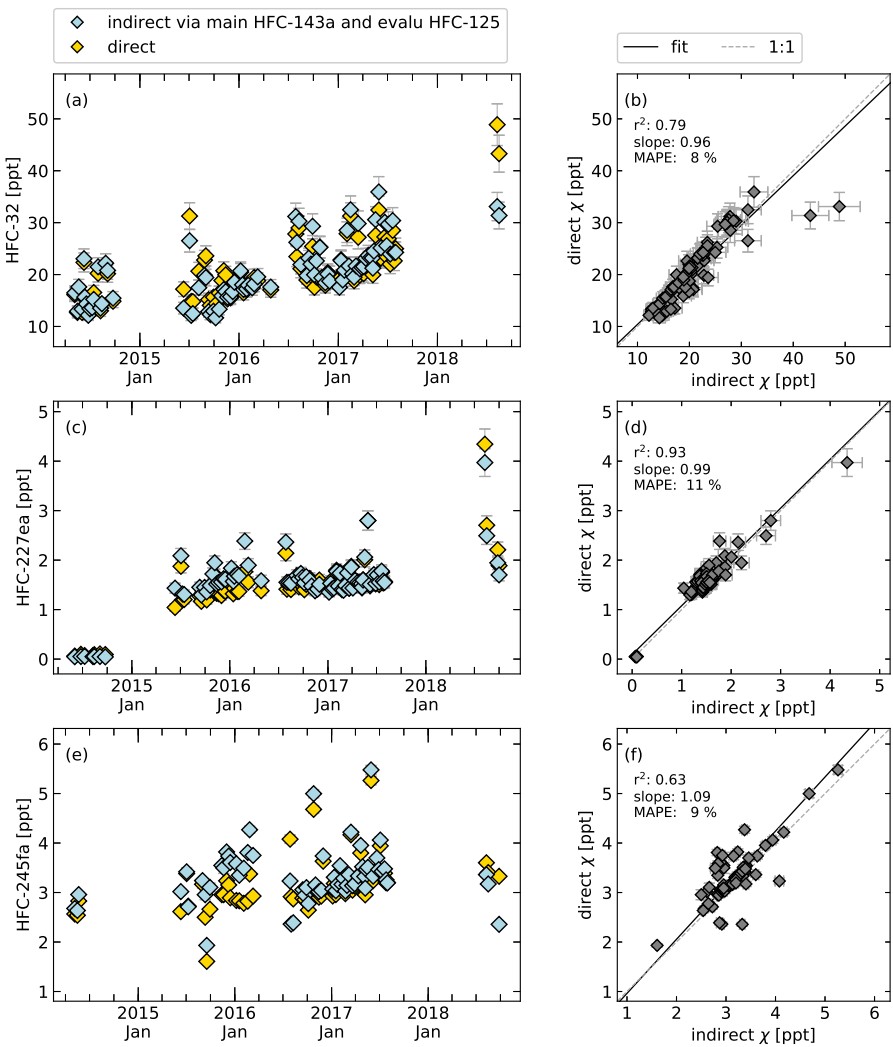

**Figure 6.** Time series (left) and correlations (right) of the mole fractions of HFC-32, HCFC-227ea, and HFC-245fa calculated directly (yellow symbols) and indirectly (light blue symbols) for the weekly flask sample measurements. HFC-143a was used as main reference substance, HFC-125 as evaluation substance to select data with constant $rRF$. Error bars, which indicate the measurement precisions, are included but are often smaller than symbol size.

### 3.2.3 Evaluation based on in situ measurements

Fig. 7 shows the results of the data selection procedure for the in situ GC-MS at Taunus Observatory, using HFC-152a (left) and HFC-245fa (right) as main reference substances for the period May 2018 through March 2019. Shown are daily mean



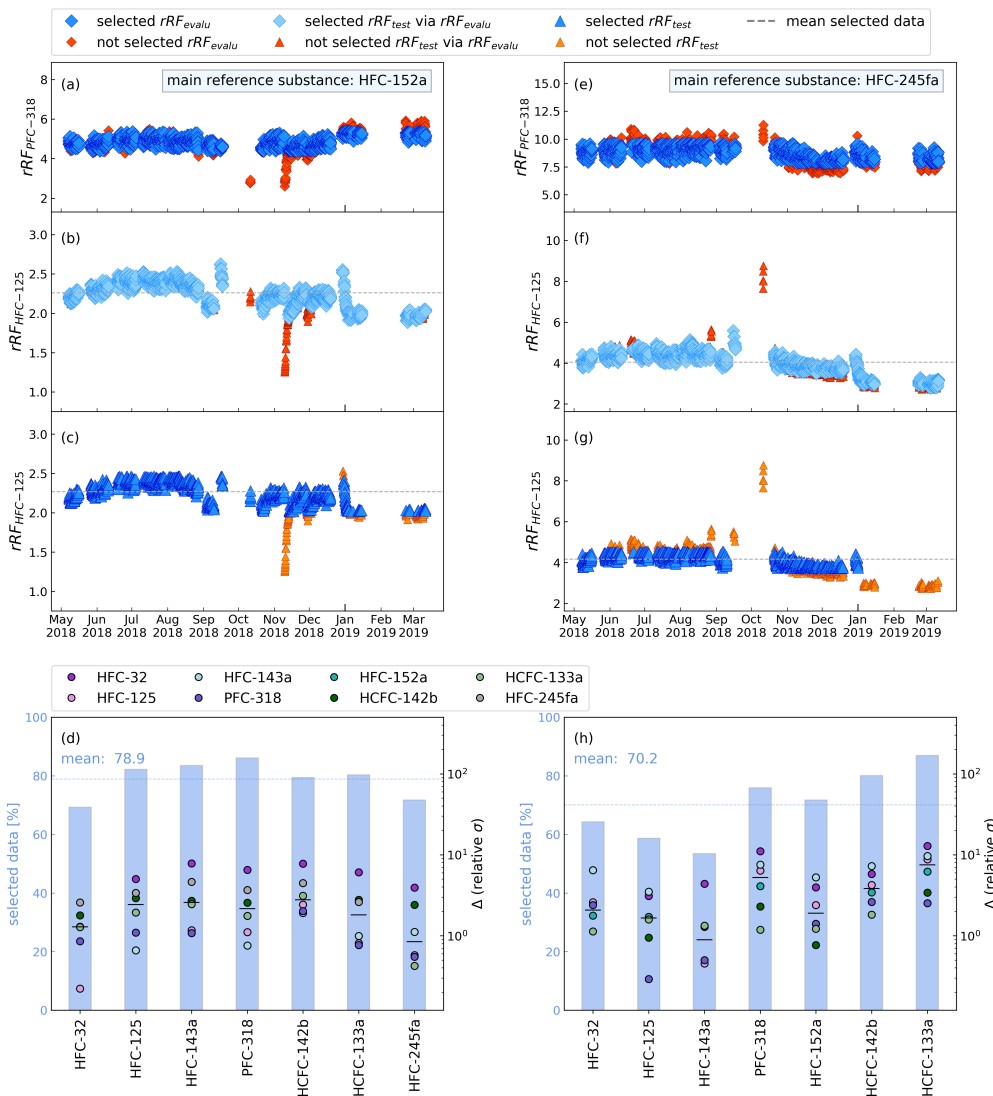

**Figure 7.** As Fig. 5 but for the continuous in situ measurements at Taunus Observatory using main reference substances HFC-152a (a-d) and HFC-245fa (e-h) and PFC-318 as evaluation substance.

values for simplicity, but two-hourly data were used for all calculations. Histograms in panels (d) and (h) show that a large percentage of data meet the filter criterion and a larger fraction of data is selected from the in situ measurements than from during the weekly flask sample measurements. HFC-152a as main reference has a high overlap within the 10 %-range with the other substances, with a mean of 78.9 % data selected. For HFC-245fa this is only 70.1 %.

Fig. 8 shows the comparison of directly and indirectly calculated relative responses and mole fractions for HFC-125, HFC-143a, and HFC-245fa. Again, daily mean values are shown for simplicity while all calculations were performed for the air





**Table 3.** Average relative differences of directly and indirectly calculated mole fractions for the in situ GC-MS measurements for the period from May 2018 to March 2019 (cf. Fig. 8). As main reference HFC-152a, as evaluation reference PFC-318 is used.

| compound | av. rel. difference | standard deviation |
|----------|--------------------|--------------------|
| HFC-125 | 18.2 % | 5.1 % |
| HFC-143a | 16.5 % | 5.9 % |
| HFC-245fa | 15.4 % | 9.8 % |
| HFO-1234yf | 24.3 % | 15.2 % |
| HFO-1234ze(E) | 19.5 % | 12.2 % |

measurements every second hour. For the continuous measurements, HFC-152a is used as main reference and PFC-318 as
evaluation substance. As was the case for the flask sample measurements, some features of the time series are caught well. Especially shorter-term variations are well captured, while long-term trends between the directly and indirectly calculated mole fractions are partly different between the directly determined and the indirectly determined data. This is caused by long-term drifts in the $rRF$. The average relative differences are given in Table 3.

## 4    Application of indirect calibration method to short-lived synthetic halocarbons

As the indirect calibration method has shown satisfactory results for the test substances, we apply it to the short-lived compounds HFO-1234yf, HFO-1234ze(E), and HCFO-1233ze(E). For these compounds, the direct calibration is limited to parts of the time series which were calibrated with gases containing these substances at sufficiently high mole fractions.

Fig. 9 shows a time series of measurements with the two GC-TOF-MS systems from January 2014 to May 2020 for the weekly whole air sampling measurements, and from May 2018 to May 2019 for the continuous in situ measurements. A com-
parison of direct and indirect calibrated data can only be performed for the continuous in situ measurements, where the calibration gas used until March 2019 contained detectable amounts of HFO-1234yf (0.149 ppt) and HFO-1234ze(E) (0.199 ppt). The average relative differences of that comparison are given in Table 3. For HFO-1234yf, the mole fractions differ by around 24.3 %, for HFO-1234ze(E) the relative average difference is 19.5 %.

Data of the weekly flask sampling (cf. Table 4) show increasing detectability for the investigated H(C)FOs since 2014, as
seen in Vollmer et al. (2015). In 2014 only HFO-1234y was detected. Until 2018 the detectability of HFO-1234yf and HCFO-1233zd(E) increases continuously up to 100 %, whereas in 2019 they were detected in 96 % and 98 % of measurements. From 2015 to 2019 HFO-1234ze(E) mostly shows a detectability of 100 %, except for 2016 (92 %) and 2019 (98 %). From 2014 to 2019 mole fractions range between 0–5 ppt (0–2 ppt calculated indirectly) for HFO-1234yf. Its annual mean mole fractions increased from 0.03 ppt in 2014 (calculated indirectly) to 0.81 ppt in 2019 (calculated directly). The high mean mole fraction for
the indirectly calculated data of HFO-1234yf in 2018 (1.23 ppt) could be due to only four samplings available between August and October, where mole fractions seem to be higher due to an annual cycle. For HFO-1234yf(E) mole fractions from 2015 to 2019 mostly range between 0–2 ppt, except for a few outliers. Annual mean mole fractions of HFO-1234ze(E) increased

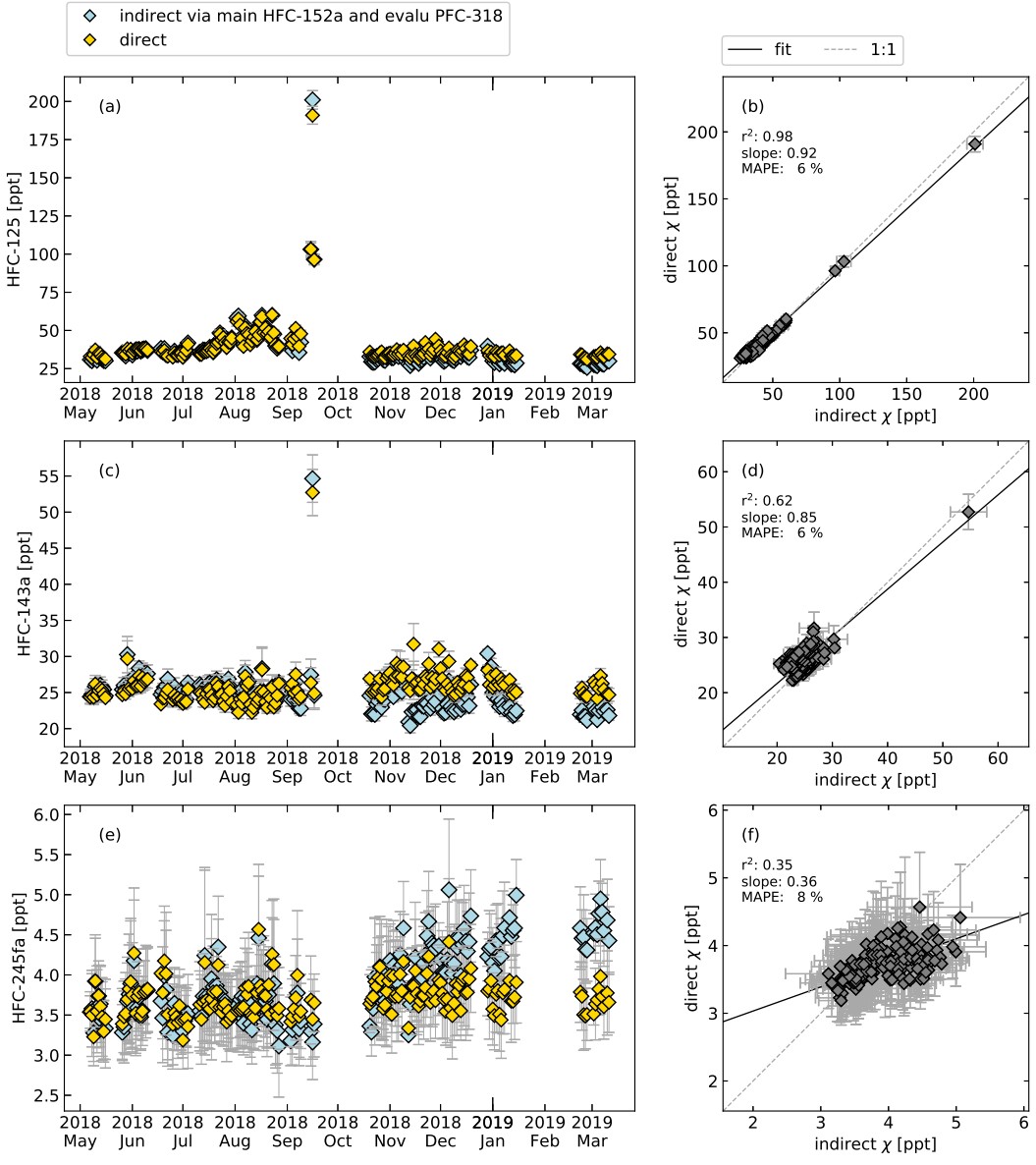

**Figure 8.** As Fig. 6, but for the continuous in situ measurements at Taunus Observatory using GC-MS and for the following compounds: HFC-125, HFC-143a, and HFC-245fa, calculated directly (yellow symbols) and indirectly (light blue symbols). Here, HFC-152a was used as main reference substance, PFC-318 as evaluation substance to select data with constant $rRF$. For reasons of simplified illustration daily means are shown. The error bars indicate the standard deviations of the measurements of one day.

from 0.12 ppt in 2015, calculated indirectly, to 1.27 ppt in 2019, calculated directly. Also H(C)FO-1233zd(E) shows increasing annual mean mole fractions, from 0.1 ppt in 2015, calculated indirectly, to 0.51 ppt in 2019, calculated directly. The annual

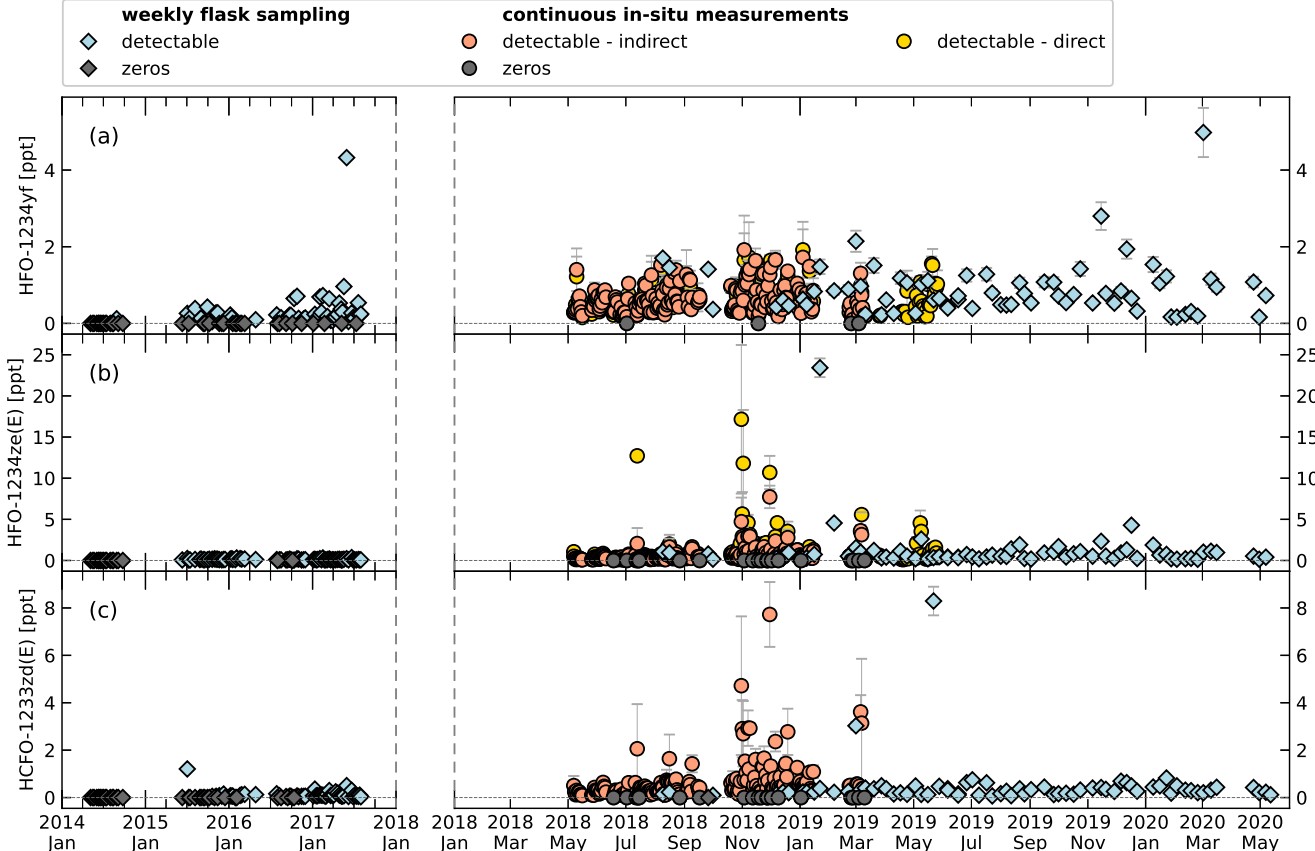

**Figure 9.** Time series of mole fractions of HFO-1234yf (a), HFO-1234ze(E) (b), and HCFO-1233zd(E) (c) at TOB. Symbol shape indicates flask sampling (diamonds) and in situ measurement (circles). Grey symbols represent mole fractions below detection limit (cf. Table 2 and 3). Data of weekly flask sampling are indirectly calibrated mole fractions before January 2018 and directly calibrated values after. For the in situ measurements, indirectly and directly calibrated mole fractions are indicated by color (orange and yellow). Note the change in x-axis scaling after January 2018 because of increased data frequency.

mean mole fractions of all three H(C)FOs at TOB are in between typical mole fractions observed at the urban Dübendorf site and the clean air site at Jungfraujoch in Switzerland (Vollmer et al. (2015) and update of Vollmer et al. (2015) (unpubl., private communication)).

Data of the in situ measurements (cf. Table 5) show a variation of detectable amounts between 96 and 98 % for HFO-1234yf and between 91 and 93 % for HF0-1234ze(E). The means of HFO-1234yf data calculated indirectly do not differ more

than 15 % from the direct calculated data (for 2019 direct and indirect). Whereas HFO-1234ze(E) shows a large deviation with maximal 44 % for the data in 2018. These larger amounts could be caused by the low amount of HFO-1234ze(E) in the calibration gas used from May 2018 to March 2019. The mean values of both substances, independently if calculated directly





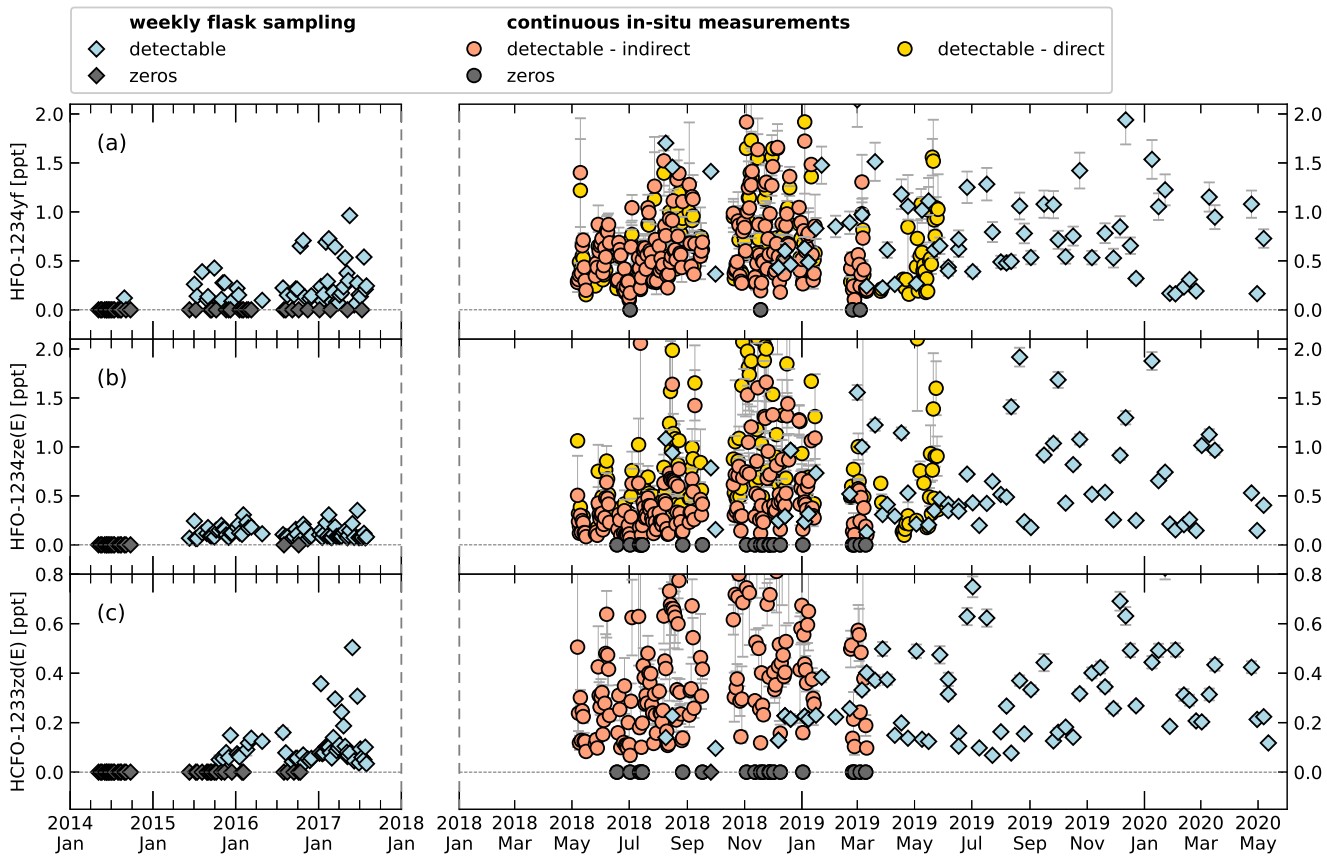

**Figure 10.** As Fig. 9 zoomed in to visualise small mole fractions better.

or indirectly, do not show an increase or decrease over the short time period of less than one year which is covered by the in situ data. The indirect mean values in 2018 for both substances compared to the indirect calculated substances using the whole

air flask sampling data are lower (mean HFO-1234yf mole fraction is 0.57 ppt lower, HFO-1234ze(E) shows a 0.11 ppt lower mean mole fraction). But as mentioned above this could be caused by an unequal time distribution of the data. This could cause also the other deviations between the in situ and whole air flask measurements. However, typical mole fractions are also in between typical mole fractions observed at the urban Dübendorf site and the clean air site at Jungfraujoch, both in Switzerland (Vollmer et al. (2015) and update of Vollmer et al. (2015) (unpubl., private communication)).





**Table 4.** Mean and median observed mole fractions (in ppt), number of observations, and the percentage of detectable peaks per year of HFO-1234yf, HFO-1234ze(E), and HCFO-1233ze(E) in the whole flask air samples. Data between 2014 and 2018 are calculated indirectly and are in italic, whereas data from October 2018 onwards are calculated directly. The indirectly estimated mole fractions are calculated using the indirect calibration approach, with HFC-143a as main reference substance and HFC-125 as evaluation substance. Mean and median values include measurements with undetectable mole fractions, as it is performed in Vollmer et al. (2015). Instead of assigning those a value of zero, values equal half of the detection limits were assigned to them. The limits of detection (LODs), which are calculated as the 1.5-fold noise of each chromatogram, are 0.04 ppt (HFO-1234yf), 0.06 ppt (HFO-1234ze(E)), and 0.05 ppt (HCFO-1233zd(E)).

[1]Data of 2018 is calculated indirectly until October 23.

[2]Data of 2018 is calculated directly since October 24, using the calibration standard cal GUF-16.

| compound | 2014 | 2015 | 2016 | 2017 | 2018[1] | 2018[2] | 2019 |
|---|---|---|---|---|---|---|---|
| | *indirect* | *indirect* | *indirect* | *indirect* | *indirect* | direct | direct |
| HFO-1234yf | | | | | | | |
| mean [ppt] | *0.03* | *0.13* | *0.14* | *0.4* | *1.23* | 0.5 | 0.81 |
| median [ppt] | *0.02* | *0.09* | *0.11* | *0.2* | *1.44* | 0.46 | 0.69 |
| numb of obs | *1* | *11* | *13* | *26* | *4* | 6 | 50 |
| % detectable | *6* | *58* | *54* | *87* | *100* | 100 | 96 |
| HFO-1234ze(E) | | | | | | | |
| mean [ppt] | — | *0.12* | *0.13* | *0.12* | *0.74* | 0.5 | 1.27 |
| median [ppt] | — | *0.12* | *0.11* | *0.09* | *0.86* | 0.29 | 0.5 |
| numb of obs | *0* | *19* | *22* | *30* | *4* | 6 | 51 |
| % detectable | *0* | *100* | *92* | *100* | *100* | 100 | 98 |
| HCFO-1233zd(E) | | | | | | | |
| mean [ppt] | — | *0.1* | *0.06* | *0.13* | *0.12* | 0.19 | 0.51 |
| median [ppt] | — | *0.03* | *0.56* | *0.09* | *0.12* | 0.21 | 0.27 |
| numb of obs | *0* | *6* | *17* | *30* | *3* | 6 | 51 |
| % detectable | *0* | *32* | *71* | *100* | *75* | 100 | 98 |

## 5 Summary and Conclusions

Non-target analysis using full-scanning mass spectrometry offers the opportunity to detect and quantify species in the atmosphere retrospectively. However, as gas chromatography is a relative measurement technique, knowledge of the mole fraction of the retrospectively analysed species in the calibration gas is required. Often the species of interest is either not detectable in the calibration gas or the mole fraction in the calibration gas is not known. For such cases we have developed an indirect calibration approach which relies on the assumption that the relative sensitivity of the analytical system to two species changes in a similar way, so that their ratio would be constant in time, even if the absolute sensitivity of the system changes. In this case, quantification may be performed using the measurement of a reference species and the ratio of the relative sensitivities





**Table 5.** As Table 4 but for the continuous in situ measurements at TOB from May 2018 to May 2019 for HFO-1234yf and HFO-1234ze(E). The indirect mole fraction estimations from May 2018 to March 2019 are based on HFC-152a as main reference substance and PFC-318 as evaluation substance. The limits of detection (LODs), which are calculated as the 1.5-fold noise of each chromatogram, are 0.02 ppt for HFO-1234yf and 0.01 ppt for HFO-1234ze(E) for the in situ measurements.

[1]Data of 2019 is calculated directly and indirectly until March 2019.

[2]Data of 2019 is calculated directly only since March 2019, using the calibration standard GUF-17.

| compound | 2018 direct | 2018 *indirect* | 2019[1] direct | 2019[1] *indirect* | 2018+2019[1] direct | 2018+2019[1] *indirect* | 2019[2] direct |
|---|---|---|---|---|---|---|---|
| HFO-1234yf | | | | | | | |
| mean [ppt] | 0.63 | *0.66* | 0.51 | *0.58* | 0.61 | *0.65* | 0.54 |
| median [ppt] | 0.54 | *0.58* | 0.41 | *0.49* | 0.53 | *0.56* | 0.44 |
| numb of obs | 1582 | *1440* | 259 | *145* | 1841 | *1585* | 303 |
| % detectable | 98 | *98* | 97 | *96* | 98 | *98* | 94 |
| HFO-1234ze(E) | | | | | | | |
| mean [ppt] | 1.11 | *0.63* | 0.82 | *0.65* | 1.07 | *0.63* | 0.65 |
| median [ppt] | 0.52 | *0.37* | 0.51 | *0.43* | 0.52 | *0.38* | 0.3 |
| numb of obs | 1474 | *1338* | 250 | *139* | 1724 | *1477* | 282 |
| % detectable | 91 | *91* | 93 | *92* | 92 | *91* | 88 |

of target and reference compound, provided that the absolute value of the relative response of the species is derived retrospectively. In order to evaluate the stability of the relative responses of two such species, we tested the approach using species
whose concentrations are known in the calibration gas. We suggest that it is useful to use an evaluation substance to select periods when relative responses of the measurement system are rather stable. Further, it is likely that using reference species with similar retention times as the target species provides more stable results. By analysing correlations and variabilities of the relative responses, we identify the combination of a reference and an evaluation substance which yields good results for a range of different target gases. Furthermore, we have chosen to include only time periods where the relative response of the reference
substance and the evaluation substance are stable within 10 % in the analysis. A good combination of reference and evaluation substance should thus yield small deviations between direct and indirect calibration for a wide range of compounds, while also retaining a sufficient number of measurements based on the filter criterion of maximum deviation in relative response factor.

This procedure with the 10 % criterion is applied to two different data sets for testing. The first data set is a measurements time series of flask samples collected at the Taunus Observatory on the *Kleiner Feldberg* near Frankfurt in Germany. This data
set has been evaluated for the time period from October 2013 to December 2018. The second data set is from in situ measurements at the Taunus Observatory using an automated gas chromatographic system with time-of-flight mass spectrometric detection. This data set has been evaluated for the time period from May 2018 to March 2019. For the long-term flask data, we find relative differences between directly and indirectly calibrated mole fractions of different gases ranging between 13





and 21 %. For the in situ data, differences between directly calibrated and indirectly calibrated mole fractions ranged between
about 15 and 25 %.

Based on these differences between directly calibrated and indirectly calibrated values of up to 25 %, we conclude that the indirect calibration method is not suited for detection of small trends of long lived gases in the atmosphere, which are often of the order of less than 1 % per year. However, for species with large trends where no direct measurements are available, this method can provide the correct order of magnitude of atmospheric mole fractions in the past. A further interesting application
is to the measurement of short-lived gases, which are excepted to show high variability in the atmosphere. For such gases, both correct orders of magnitude and also the frequency at which they are observable can be derived. In order to confirm the validity of the indirect calibration approach, it will be useful to maintain aliquots of calibration gases, so that these can be calibrated retrospectively allowing to confirm the stability of relative response factors for species which are detectable and stable in the calibration gas over a a longer time period.

Examples for species where the indirect calibration is useful are the unsaturated HFOs and HCFOs, which have recently been introduced as replacement compounds for long-lived hydrofluorocarbons. These gases are short lived with local lifetimes of less than a month and are increasingly used for e.g. mobile air conditioning. The three H(C)FOs HFO-1234yf, HFO-1234ze(E) and HCFO-1233zd(E) have been detectable at an increasing frequency in our ambient air chromatograms. We have thus applied the indirect calibration method to both the flask measurements and the in situ measurements of H(C)FOs. For
the flask measurements we show that the frequency at which measurable peaks are observed at Taunus Observatory increases with time. All H(C)FOs are present in nearly all flask samples collected at the Taunus Observatory since early 2018, while samples from the year 2014 only showed very occasional measurable concentrations of HFO-1234yf. Consequently, typical mole fractions increased from below 0.1 ppt in 2014 (indirectly calibrated) to median values between 0.25 ppt for HCFO-1233zd(E) and 0.7 ppt for HFO-1234yf in 2019, based on a direct calibration. While the direct calibration is also preferable,
the indirect calibration offers additional useful information in this case. This observed increase in the mole fractions and the frequency of observations is in line with the observations by (Vollmer et al. (2015) and update of Vollmer et al. (2015) (unpubl., private communication)) at the remote station of Jungfraujoch. As expected, the mole fractions observed at Taunus Observatory are in between those reported for the remote station of Jungfraujoch and the urban station of Dübendorf in Switzerland (Vollmer et al. (2015) and update of Vollmer et al. (2015) (unpubl., private communication)).

*Data availability.*  Data are available from the corresponding author upon individual request.

*Author contributions.*  Method development and testing was performed by Markus Jesswein, Fides Lefrancois, Andreas Engel, and Tanja Schuck. GC-MS measurements in the laboratory and at site were performed by Fides Lefrancois, Markus Thoma, and Tanja Schuck. The manuscript was prepared by Fides Lefrancois, Andreas Engel, and Tanja Schuck. All authors contributed to the discussion of results.





*Competing interests.* The authors declare that they have no conflict of interest.

*Acknowledgements.* The authors acknowledge the contribution of technical staff at the Institute for Atmospheric and Environmental Sciences and Taunus Observatory and all staff involved in regular sample collection, in particular Frank Engel, Timo Keber, Robert Sitals, and Kieran Stanley. We also would like to thank Martin Vollmer (EMPA) for calibration of laboratory standards.



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
