# Peer review of "An indirect-calibration method for non-target quantification of trace gases applied to a time series of fourth generation synthetic halocarbons at the Taunus Observatory (Germany)"

_Atmospheric Measurement Techniques, 2020_

## Author Comment (AC1)

**Reply to RC 1**

**Manuscript information:**

- Title: Non-target analysis using a gas chromatography with time-of-flight mass spectrometry: application to time series of fourth generation synthetic halocarbons at Taunus Observatory (Germany).
- Authors: Fides Lefrancois, Markus Jesswein, Markus Thoma, Andreas Engel, Kieran Stanley, and Tanja Schuck
- MS No.: amt-2020-488
- MS type: Research article
- Iteration: Final response (AMT Discussions)

We would like to thank the reviewer for the constructive and detailed comments.

**Point-by-Point reply:**

1. *"The analysis raised a question for me (but that doesn't need to be addressed in the manuscript). The question is why don't all compounds work equally well as internal standards? And if they don't all work well, what is the uncertainty of the non-target compounds for any specified period? They might behave well or might not, it seems to me. Also, given the stated precisions of the calibration standards, it is then surprising that the correlations between standard responses show percent errors in the 10% range. I'm not sure how to interpret that."*

We do not exactly know what effects the different substances and why they do not behave the same way in different standards or measurements. Some substances could be influenced by changing $H_2O$-amounts or different ionization processes. We investigated several parameters of the compounds (e.g. retention times, signal intensities, compound similarities) but we could not find any conclusion yet. The internal calibration standard precision does not have an influence on this approach. Finally, we have to say, it is also a system dependent issue. We tried to point this more in section 3.2.1 *Relative Response Factor* and answer on point 5.

2. *L 81. Suggest eliminate "preceding" or change to "…(ppt)), sample trace gases are enriched by cryofocussing in a sample loop."*

We changed it to: "(…), a cryofocussing sample loop unit is used to enrich the trace gases (Obersteiner et al., 2016b)."

> ibration standard. Due to the low mole fractions of the investigated substances (range of picomole per mole; pmol mol$^{-1}$,
> 95   or hereafter, parts per trillion ; ppt), a cryofocussing sample loop unit is used to
> enrich the trace gases (Obersteiner et al., 2016b). The sample loop, a 1/16" stainless steel tube of 10 cm length, is filled with

3. *L113. Though I think I know what you mean, could you better describe what a "target" standard is?*

We added a comment on that:

> hours. Following every 13th air measurement, a target standard  gas is measured. The target gas is a cylinder of known concentration, which is measured regularly on the system to monitor the
>
> 140    stability, especially possible drifts in the calibration gas. From May 2018 to March 2019 the calibration gas used was a whole

4. *L120. It would be helpful to describe some more detail of the method. For example, I couldn't find anywhere how the quantitation was accomplished with the QTOF data. Was a single selected ion used for each compound, or the sum of major fragments, or some integration of a peak that has been deconvolved from the total ion chromatogram? Were different methods of sample integration tried? Perhaps there is also a relationship between mass and total ion current that could be used to quantitate certain classes of compounds (at least within 25%)? Other questions: mass resolution of TOF?*

We have followed the reviewer's suggestion and added some more detail on the quantification to the manuscript:

> **2.4   Data evaluation**
>
> 145    For both measurement setups, the integration of the chromatographic peaks is performed in a similar way as described in Schuck et al. (2018). For the quantification of individual substances we used single ions. These ions were chosen in previous analyses, in order to avoid overlap with ion fragments from possibly co-eluting substances and at the same time provide high signal to noise ratios. The signal areas $(A)$ of each substance are divided by the enriched sample Volume $(V)$ to yield a response

The mass resolutions of the used TOFMS systems are given in the quoted manuscripts in section 2 *measurements*.

5. *L177. Although HCFC_141b elutes near water, it shows excellent precision. So not sure why this might be excluded. Or it might be interesting to learn how water vapor might influence the results.*

We added a comment on that:

> 210    trifluoroethane, $CCl_2FCClF_2$, CAS 76-13-1) in the case of the laboratory system. In these cases, water influences the signal intensity of the two compounds in the analysis in the laboratory system. Comparing them to their own intensities, as it is used in the direct calibrated analysis, they still show the mentioned precision. Due to the indirect calibration method, this change in signal intensity leads to an incomparability with other compounds not influenced by water vapor.

**6.** *L179. Not sure if the plots are artificial data from some "arbitrary substances" or if the actual compounds are just not named here. Could you clarify?*

The figure is a „dummy"-data-set, to explain the methodology schematically. We added a clarification in the text.

> Fig. 1 shows a schematic example for the identification of periods of stable $rRF$, where a random dataset was created.
> 215    Panel (a) shows the $rRF_{evalu}$ for two arbitrary substances, a so called main reference and an evaluation substance, which

> **Figure 1.** Schematic example using a random dataset of the identification of periods with constant $rRF$ for an undetectable substance in the calibration standard. Panels (a) and (b) show the calculated $rRF_{evalu}$ of a known main reference and a known evaluation substance. Panel (b) shows which measurements will be selected, excluding measurements where the $rRF$ differs more than 10 %. The resulting selection of measurements should represent the periods of stable $rRF_{test}$ in panel (c) and (d), where the $rRF$ is determined using the main reference substance and an arbitrary test substance. The aim is to find a main reference and an evaluation substance, which have many measurements with a constant $rRF$ and which will represent the selection of test substances as well as possible.

**7.** *L182. It is not clear how the 10% criterion for rejection is applied. Is this from point to point, or relative to some average?*

We clarified this by changing the manuscript as follows:

> are both detectable in the used calibration standard. To identify periods of stable $rRF_{evalu}$, for each individual measurement the number of  measurements with an $rRF_{evalu}$ that differs by not more than 10 % is counted.  Therefore, every single data point was compared to all other data points iteratively. The data point with the highest number of matching data points is used as a reference and all measurements that fall outside the 10 % interval are excluded (shown
> 220    as grey data points in panel (b)). If more than one measurement has the same number of matching data points, the case with

**8.** *L197. Not sure if you mean "exemplary", as in "best example of the group", or are these just examples of some of the compounds. (also in Figure caption).*

These are illustrative examples of some of the investigated substances to demonstrate that some substances correlate well, some not. We clarified this as follows:

> are both calculated for all calibration gas measurements during the measurement routines of the air samples. Fig. 2 shows
> 240     illustrative correlations of peak areas for HFC-143a (a), HFC-125 (b), and HFC-227ea (c), versus HFC-152a, PFC-318, and HCFC-133a. Except HFC-227ea (column (c)), the presented substances and their comparisons of

> **Figure 2.** Correlation of peak areas of  illustrative substances from calibration gas measurements of phase where calibration cylinder GUF-10 was used, their coefficient of determnination ($r^2$) and the mean absolute percentage error (MAPE). Shown are the substances HFC-143a in column (a), HFC-125 in column (b), and HFC-227ea in column (c) and their comparison to HFC-152a (first row), PFC-318 (second row), and HCFC-133a (third row).

**9.** *Figure 2. As I understand it, this figure compares the peak areas of compound pairs in the same calibration standard over the time of the study. Could you comment on the very large range of peak areas that were observed? Is this a characteristic of the TOF?*

We do not believe that this is a special characteristic of the TOF. It should be noted that measurements cover a period of nearly five years, where the sensitivity can change due to e.g. degradation of filaments and detectors, tuning of the instrument etc. We have added a comment on that in the manuscript:

>  Even if the observed substances show a wide range of peak areas, it has to be mentioned that they mostly correlate well, while the
> 245 observed time period covers nearly five years, where system sensitivities has been changed over time. To test which pairs of substances produce the highest correlations, all possible pairs of substances were tested. The obtained values for $r^2$ and

**10.** *L204. Note that the independence of rRF will also depend on linearity and any zero offset.*

This is true, we mention this now in the manuscript:

> As the $rRF$ is referenced relative to the mole fraction of the measured gas, this value should be independent of the mole
> 250 fractions and thus should also remain constant after a change of standard, depending on the linearity and any zero off set.

**11.** *L209. The observed shift of 152a relative to 133a deserves some comment. Presumably there was no similar shift in the time series ambient measurements of either gas. So, this behavior, though maybe rare, would seem to be a major red flag in applying the proposed method with confidence.*

The reviewer is correct in pointing out that this is a critical point. We added a note of caution explaining that in some cases such outliers may occur, which may not be caught by the preprocessed data analysis and its filtering method.

> change of standard as a dashed vertical line. While for most combinations  the $rRF$  does not show a systematic change,
> the $rRF$ of HCFC-133a relative to HFC-152a shows a significant shift. However, this shift in $rRF$ started before the change
> 255 of standard and is thus obviously not related to an inconsistent calibration in the two standard gases used. The reason for this  shift is not known, but this is illustrative of the limitations of the indirect calibration method. Under such extreme cases, strong shifts would be observed in the atmospheric measurements and such shifts should thus be treated with care. For

**12.** *L213. There are a number of compounds that have drifts or anomalies that prevent them from being used as "reference" compounds. Does this have any implication on how these are used for direct calibration? Do these standards cause the sample mixing ratios to be flagged? The authors also suggest that there are a number of potential factors that will influence the relative responses. In cases of outliers or large shifts (such as 152a), have the authors determined specific causes for the deviations?*

The drift in relative sensitivities between two compounds does not have any direct implication for the direct calibration. So, there is no need to flag these substances in the direct calibration. As mentioned above, we investigated several potential reasons for the deviations (differences between the retention times, different signal to noises/signal intensities) of this occurrence. But – using these preselected substances – we could not find a regularity. This could be an interesting investigation in further studies, but we prefer not to elaborate on this in the present study.

**13.** *L214. Not sure of meaning...change "suited" to "suitable"?*

Changed that.

**14.** *L251. I was interested to see that 152a was selected as a reference standard for the in-situ measurement evaluation, though there was a problem with this compound in the canister analysis. As noted, this is disturbing and deserves comment.*

We added a comment on that:
* * *
**3.2.3 Evaluation based on in situ measurements**

310  For the application on the continuous in situ measurements, the preselection of main reference and evaluation substances yields different results. This implies a strong system dependency of the method and a need to evaluate appropriate substances for indirect calibration per system. In our case we can observe such a different behaviour in substance selection for HFC-152a. While it is not applicable for the indirect calibration method (cf. Fig. 4), but using the in situ measurement set up it is our best selection within the training dataset. Fig. 7 shows the results of the data selection procedure for the in situ GC-MS at Taunus

---

## Author Comment (AC2)

**Reply to RC 2**

Manuscript information:

- Title: Non-target analysis using a gas chromatography with time-of-flight mass spectrometry: application to time series of fourth generation synthetic halocarbons at Taunus Observatory (Germany).
- Authors: Fides Lefrancois, Markus Jesswein, Markus Thoma, Andreas Engel, Kieran Stanley, and Tanja Schuck
- MS No.: amt-2020-488
- MS type: Research article
- Iteration: Final response (AMT Discussions)

We would like to thank the reviewer for the constructive and detailed comments.

**Point-by-Point reply:**

1. *General comment: The manuscript contains a large number of figures and perhaps one or two of those could be relegated to SI ?*

We have gone through the manuscript and considered reducing the number of figures. However, we found that all figures are necessary to follow our evaluation approaches and therefore we prefer not to reduce the number of figures.

2. *Abstract – line 7: "thus" can be omitted in the sentence.*

Done.

3. *p. 2 line 36: Typo in spectrometry (spectrometry).*

Done.

**4.** *p.2, line 53: It is incorrect to say "TFA is known to cause negative environmental impacts". Large concentrations of TFA will do that, but the authors should consult the references they cite themselves, especially Solomon et al., for a precise characteristic on this matter, what impact the current and predicted future levels of TFA in the environment will actually have.*

Changed this as follows:

> as the main breakdown product in the atmosphere (Burkholder et al., 2015). TFA
> 60  can accumulate in water and soil and can become moderately toxic to organisms (Ellis et al., 2001;
> Russell et al., 2012; Solomon et al., 2016). In recent studies it seems the TFA amount formed from the mentioned substances
> in the troposphere may be too low to cause negative effects on human health (Solomon et al., 2016). But there is a necessity,
> to investigate these sources of TFA in more detail, as it is done in Solomon et al. (2016) or Freeling et al. (2020). Our data and
> approach can be a helpful additional tool for those investigations, and the exploration of seasonality, or temporal and spatial
> 65 trends.

**5.** *P4. line 115. Mole should be capitalized ion the beginning of the sentence.*

Done.

**6.** *p.6 line 159-161: This sentence should be rewritten for clarity. It would benefit form some commas and perhaps start out with "Using equating 3, the rRF for the species of interest, which is not.....*

Changed as suggested:

> calibration method.  Using Eq. (3), the $rRF$ for the species of interest which is not present in the standard, relative to a
> 190 compound which is detectable in the standard, can then be derived from measurements of another sample which has detectable
> amounts and known mole fractions of both species

**7.** *p.6, line 165: replace "should" with "is assumed to"*

Done.

**8.** *p.7, line 173: I guess the selected compounds listed in table 1 could be termed "a training" set for the method. As such, the authors, and other researchers, will adopt similar or dissimilar training sets, for the technique to be "calibrated' on for application to their datasets.*

We thank the reviewer for this useful suggestion. We added the following:

> **Table 1.** System precision ($1\sigma$) of the investigated substances treated as a training set of the TOF-MS used for the weekly whole air sampling
> (prc (TOF_Lab)) and of the TOF-MS used for the in situ measurements (prc (TOF_in situ)) and their calibration scales.

> of the $rRF$ are and to determine periods of low variability of the $rRF$, we have investigated the temporal change of $rRF$
> 205 for the combination of selected compounds listed in Table 1, which we call a training set here. Substances in Table 1 were

**9.** *p.7, line 182: 10% - this this value arbitrarily chosen? Why not based on a statistical parameter such as sigma(s)?*

Yes, it is arbitrary chosen. We decided to use this 10 % criterion instead of sigma(s). But it could be adjusted in further investigations. We added a comment on that:

> the lowest standard deviation is selected. In our application, we have arbitrarily chosen a maximum deviation of 10 % as a selection criterion, as it allowed the retention a sufficient number of measurements while still eliminating data which would have particularly large uncertainty. Depending on the stability of the instrument and the desired results, different criteria could be chosen. Allowing for larger deviations would result in retaining more data with larger uncertainties, while applying a more
> 225 stringent criterion would result in a dataset with less data yet likely also lower uncertainties. In panels (c) and (d) the evaluation

**10.** *Figure 4 caption: replace "used cylinders" with "used calibration gas". All figures could in general benefit from being made color "agnostic". Several of them, e.g. fig 6 and 8, are only legible in color print out, whereas there are no limitations on symbols shape that dictates the necessity of using colors.*

Changed wording as suggested and we changed the colors and symbols of Figures 6 and 8 to follow the suggestion of the reviewer. They are now more "agnostic", yet are still best viewed with color options.

[Figure]

**Figure 6.** Time series (left) and correlations (right) of the mole fractions of HFC-32, HCFC-227ea, and HFC-245fa calculated directly (yellow symbols) and indirectly ( blue symbols) for the weekly flask sample measurements. HFC-143a was used as main reference substance, HFC-125 as evaluation substance to select data with constant $rRF$. Error bars, which indicate the measurement precisions, are included but are often smaller than symbol size.

[Figure]

**Figure 8.** As Fig. 6, but for the continuous in situ measurements at Taunus Observatory using GC-MS and for the following compounds: HFC-125, HFC-143a, and HFC-245fa, calculated directly (yellow symbols) and indirectly ( blue symbols). Here, HFC-152a was used as main reference substance, PFC-318 as evaluation substance to select data with constant $rRF$. For reasons of simplified illustration daily means are shown. The error bars indicate the standard deviations of the measurements of one day and thus reflect the daily atmospheric variability rather and do not include systematic errors due to the indirect calibration method.

**11.** *P.15, line 254-255: Please comment on the fact that a larger fraction of data from the in-situ measurements than from the flask measurements were selected. Was this expected? Any predictable reason behind this outcome?*

We added a comment on that.

from the in situ measurements than from during the weekly flask sample measurements. This could be due to the shorter time period covered by the in situ data and also due to the continuous measurements in contrast to flask measurements in the

320    laboratory, where the instrument is in standby for longer time periods. HFC-152a as main reference has a high overlap within

**12.** *P.16, line 262: What does "partly different" mean here? Could a different description be used?*

We clarified that:

Especially shorter-term variations are well captured, while long-term trends between the directly and indirectly calculated mole fractions are partly different sometimes show systematic differences between the directly determined and the indirectly determined dataand indirectly determined methods. This is caused by long-term drifts in the $rRF$ and shows clearly that the indirect calibration measurement should only be applied when investigating very large long-term trends when no directly

330    calibrated measurements are available. The average relative differences are given in Table 3.

**13.** *p. 16 line 275, Suggest replacing detectability with detection frequency.*

Changed that as suggested.

**14.** *p.16, line 278 and 281: inset " weekly" before "mole fractions".*

Added that.

**15.** *p.18, line 285-287: How would this indirect retrospective method likely work out trained on a data set like that collected at Jungfraujoch – i.e., a setting with largely clean background air? Any ideas if it would work there as well?*

We think that the method would work out probably similar. We have to mention, that the long term detection of minor and small changes is not the strength and not the aim of this method. We decide not to change the manuscript to point that out. We added a comment on that few lines before (cf. answer on point 12).

16. *p. 19, line 295-297: Does this mean that indirect values obtained from datasets involving flask measurements and in-situ measurements, respectively, shouldn't really be directly compared? I.e., if the variability is quite unpredictable, which sample dataset is "generally" more likely to produce good indirect values.*

No, this does not mean that such data sets should not be combined. It is just again due to the possibility of long-term systematic trends in rRF. We added a note on this:

> 365    unequal time distribution of the data.  Also, this could cause the other deviations between the in situ and
> whole air flask measurements. Such deviations are to be expected due to possible long-term drifts in rRF, again emphasising
> the point that the indirect calibration method is better suited to investigate short-term variability in ambient air measurements
> than for the detection of long-term trends. However, typical mole fractions are also in between typical mole fractions observed

17. *p.21, line 311: "sufficiently" is probably a better word here than "rather".*

Changed as suggested.

18. *p.21, line 311-312: regarding "reference species with similar retention times" – has the sensitivity to the retention-times been tested? Is it just assumed "likely"- I'm not saying that this is not a good assumption , just wondering if the authors have tested this – if not, this should be an easy test within the GC-TOFMS data sets.*

We added a comment on that:

> using species  with concentrations that are known in the calibration gas. We suggest that it is useful to
> use an evaluation substance to select periods when relative responses of the measurement system are rather stable. Further,
> it is likely that using reference species with similar retention times as the target species provides more stable results, which
> should be investigated with a larger number of training substances. The training dataset used in this work could not confirm
> 385    that. By analysing correlations and variabilities of the relative responses, we identify the combination of a main reference and

19. *p.21, line 317: "... retainage a sufficient number of measurements". Sufficient here means a very low number? Ref. Table 4 where e.g. 2018 has 3-6 observations?*

We clarified that as following:

> and the evaluation substance are stable within 10 % in the analysis. A good combination of reference and evaluation substance
> should thus yield small deviations between direct and indirect calibration for a wide range of compounds, while also retaining
> 390    a  maximal fraction of measurements based on the filter criterion of maximum deviation in relative response
> factor, if possible.

**20.** *p.22, line 325: These quoted different are much lower than what shows up for the annualized values in the tables. Is the large discrepancies for the annualized values not an issues since those are what are often cited?*

In the summary, we give the averaged deviation by comparing the data points calculated directly and indirectly of each individual measurement, as shown in tables 2 and 3. Larger annual discrepancies only occur in the comparison of HFO-1234ze(E) in table 5, e.g. for 2018 annual mean direct: 1.11 ppt, annual mean indirect: 0.63 ppt. Whereas HFO-1234yf shows annual deviations between 5 and 13 % (e.g. for 2018 direct: 0.63 ppt, indirect: 0.66 ppt).

We added the text:

>  GC system with TOF-MS detection. This  dataset has been evaluated for the time period from May 2018 to March 2019. Comparing the data points of each measurement, calculated directly and indirectly, we found the following averaged relative differences of mole fractions for the investigated substances: For the long-term flask data, we find relative differences between directly and indirectly calibrated mole fractions of different gases ranging between
> 400    8 and 11 %. For the in situ data, differences between directly calibrated and indirectly calibrated mole fractions ranged between about 6 and 23 %.

---

## Author Comment (AC3)

**Reply to RC 3**

Manuscript information:

- Title: Non-target analysis using a gas chromatography with time-of-flight mass spectrometry: application to time series of fourth generation synthetic halocarbons at Taunus Observatory (Germany).
- Authors: Fides Lefrancois, Markus Jesswein, Markus Thoma, Andreas Engel, Kieran Stanley, and Tanja Schuck
- MS No.: amt-2020-488
- MS type: Research article
- Iteration: Final response (AMT Discussions)

We would like to thank the reviewer for the constructive and detailed comments.

**Point-by-Point reply:**

1. *I would suggest to use a title that contains the main purpose of the article, which is to make non-target screening quantitative. Usually, articles with "non-target screening" in their title contain discovery of new substances, which is not the scope here. Suggestions for title: "Quantitative non-target ...."*

We changed the title as follows:

"An indirect-calibration method for non-target quantification applied to time series of fourth generation synthetic halocarbons at Taunus Observatory (Germany) using gas chromatography mass spectrometry measurements."

>  An indirect-calibration method for non-target quantification of trace gases applied to a time series of fourth generation synthetic halocarbons at the Taunus Observatory (Germany)

2. *l. 8: I would simplify the grammar: "This archive can be used if", or even "This archive can be used for retrospective screening"*

Done, as suggested in the first sentence.

3. *l. 11: "or the amounts in the calibration gas may not have been quantified."*

Done.

4. *Introduction, l. 18-20: "The application of the indirect calibration method on several test cases can result into accuracies around 13% to 20 %. For H(C)FOs accuracies up to 25% are 20 achieved." I would be good to reformulate these sentences to really convey the meaning that low values represent a better accuracy. Maybe you can replace the word "accuracy" by "uncertainty" here: "The application of the indirect calibration method on several test cases can result into uncertainties around 13% to 20 %. For H(C)FOs, of particularly low mole fraction values, uncertainties up to 25% are observed.".*

Changed as suggested by the reviewer.

5. *l. 26 "which are part of or affiliated to" l.31, maybe: "is not well covered"*

Done.

6. *l. 33: "by the installation"*

Done.

7. *l. 35: maybe you want to specify the mass range coverage of the TOF instrument (minimum and maximum measured masses).*

The mass range coverage was added.

> trometry (Hoker et al., 2015; Obersteiner et al., 2016b). The TOF-MS used for the weekly whole air flask samplings scans the mass range from 45 to 500 u, whereas the TOF-MS used for the in situ measurements scans a mass range from 19 to 300 u. In addition to TOF-MS, which is  acquiring a continuous mass spectrum over the complete chromatogram, flask-air
> 40  samples are quantified using quadrupole mass spectrometry, where predefined masses are scanned at selected

8. *l. 45-47: may you can consider if you would like to leave out "HFC-1234yf" and "HFC-1234ze(E)". The HFC nomenclature is actually not made for compounds with a double bond.*

Because HFC-1234yf and HFC-1234ze(E) have been widely used in the scientific literature, we leave them here among the list of possible substance names. To make clear that these substances do not fully represent the class of HFOs we modified the following sentence:

> 50  HFC-1234ze(E), CAS 29118-24-9), and HCFO-1233zd(E) (E-1-chloro-3,3,3-trifluoroprop-1-ene, trans-CF$_3$( HCFC-1233zd(E), CAS 102687-65-0).  In the following we will use the H(C)FO nomenclature for the hydro(chloro-)fluoroolefines, as the HFC-nomenclature is not made for compounds with a double bond. These H(C)FOs are examples of

**9.** *l.51 : "have no ODP": "have an ODP value of zero." "have no ODP" suggests that the computation of the ODP is impossible.*

Changed as suggested.

**10.** *l. 57: the magnitude of what? Amplitude of annual cycles, magnitude of mole fraction of pollution events?*

Changed wording to:

> The three H(C)FOs were observed in the atmosphere for the first time around 2010–2014 at Jungfraujoch and Dübendorf in Switzerland (Vollmer et al., 2015a). The percentage of detectable mole fractions, the yearly  mole fraction and maximum mole fractions of pollution events increased at both sites after 2010, with the high
> 70    mountain site Jungfraujoch generally experiencing lower mole fractions. Vollmer et al. (2015a) identified the Benelux region

**11.** *l. 80: "and each pair of measurements is bracketed"*

Done.

**12.** *l. 81: "range of parts per trillion (ppt)": range of picomole per mole, pmol/mol or hereafter part per trillion (ppt)"*

Done.

**13.** *l. 102: add comma: "For each measurement, approximately"*

Done.

**14.** *l. 117: you can leave out the sentence about calibration scales, it is already mentioned l. 92- 94.*

Changed to:

> 140    stability, especially possible drifts in the calibration gas. From May 2018 to March 2019 the calibration gas used was a whole air standard filled in February 2015 at TOB (GUF-14). In March 2019 it was changed for a newer standard also filled at TOB in April 2018 (GUF-17).  Mole fractions of both working standards were calibrated  as described above.

**15.** *l. 139 : "before calibration standards containing measurable amounts of these substances were used".*

Done.

**16.** *l. 140: tense concordance, not sure, check with native speaker. "When these compounds were detectable in ambient air, the peak areas could not be converted to mole fractions using Eq. 2 because neither numeric values for Acal nor rR were available."*

Changed to:

> these substances.  When these compounds became detectable in ambient air, the peak
> areas  could not be converted to mole fractions using Eq.  2, because neither numeric values
> 170  for $A_{cal}$ nor $rR$  were available. Therefore, a mathematical  relationship between a compound which is

**17.** *l. 141: you surely mean: "between another compound which is measurable in the standard"*

The reviewer is correct. Changed as suggested.

**18.** *l. 144: "that means that the ratio of signal per amount of analyte for the two compounds is constant with time." I'm not sure about the meaning of this sentence. We know that the response of a MS instrument may vary strongly over time, for example the instrument response increases after source cleaning. However what is important here is that the instrument response behaviour should vary similarly over time for all substances, as you clearly write afterwards. I would rephrase as: "Ideally, the sensitivity of the analytical system for two different species should behave similarly over time. In such a case, the ratio of responses R of two given species should be close to constant."*

Changed that as suggested.

**19.** *l. 146: "this ratio should be the same for any sample.": maybe too general. Suggestion to write more specifically: "this ratio should be constant over time for any chosen pair of compounds".*

Changed as follows:

> for the two compounds  being constant in time. In such a case, the ratio of responses $R$ of
> two  given species should be close to constant. In case of equal amounts of sample ($V_{cal} = V_{air}$), the ratio can also be
> 175  computed from the ratio of the signal areas ($A$). If the responses and areas are further normalised to the mole fractions of the
> two species, this ratio should be  constant over time for any chosen pair of compounds for any sample. We

**20.** *l. 155: "It must be stable over time". Check entire manuscript.*

Changed as suggested.

**21.** *l. 164-166: meaning not clear. A non-stable sensitivity does not necessarily imply a non-stable relative sensitivity, this is something you are going to investigate next. Suggestion to rephrase: "The methodology outlined in 3.1 is based on the assumption of a constant rRF in Eq. 4. In reality, the absolute sensitivity of a mass spectrometer is known to vary over time, in particular after tuning the mass spectrometer or after modifications of the analytical system such as replacement of filaments, columns or sample loops. It is therefore an open question whether changes in the relative sensitivity rRF should also be expected or not. Thus, to evaluate [...]"*

Changed as suggested.

**22.** *l. 169: "need to separated": "need to be evaluated separately".*

Periods of stable/unstable rRF are not exactly evaluated separately. In fact, periods with unstable rRF are excluded from further analysis. We reworded the sentence correspondingly:

> analytical system such as replacement of filaments, columns or sample loops . It
> is therefore an open question whether changes in the relative sensitivity, $rRF$ . should also be expected or
> 200  not. Thus, to evaluate the approach described above, the temporal stability of the $rRF$ needs to be investigated and only periods
> with stable   are included in further analysis. In the following we will refer to the compound

**23.** *l. 181-185: difficult to understand, suggestion to rephrase: "To identify periods of stable rRFevalu for a given pair of compounds, timeseries of rRFevalu are reviewed. To do so, for each measurement or data point of rRFevalu in the timeseries, we compute the sum of other rRFevalu data points that do not deviate from the chosen data point by more than 10%. The data point with the highest number of matching data points is used as a reference (shown with red cicle in Figure 1, panel (b)) and all data points that fall outside the 10% interval are excluded (shown as grey data points in panel (b)).*

*Note: I would not use "independant measurement", since the measuring instrument is the same of course the results are not fully statistically independent, and we actually need the results not to be independent for this method to work.*

*To make it more clear, on Fig. 1 please mark with e.g. a red circle the data point that was selected as most likely rRF value.*

We agree that the measurements are not fully independent and modify the wording accordingly. With regards to Figure 6, we fear that the suggested marking of a single data point would be confusing. All data points are compared step by step to all others and this iterative procedure would not become clear by indicating the selected data points. To make the procedure more clear we added the following statement in the iterative comparison of all data points:

> the number of  measurements with an $rRF_{evalu}$ that differs by not more than $10\%$ is counted.
> Therefore, every single data point was compared to all other data points iteratively. The data point with the highest number
> of matching data points is used as a reference and all measurements that fall outside the $10\%$ interval are excluded (shown
> 220  as grey data points in panel (b)). If more than one measurement has the same number of matching data points, the case with

*24. Table 1: add bibliographic reference to all scales where needed.*

*METAS-2017: Guillevic et al., 2018 (ok, already done).*

*EMPA-2013: for HCFC-133a: Vollmer, M. K., Rigby, M., Laube, J. C., Henne, S., Rhee, T. S., Gooch, L. J., Wenger, A., Young, D., Steele, L. P., Langenfelds, R. L., et al. (2015), Abrupt reversal in emissions and atmospheric abundance of HCFC-133a (CF3CH2Cl),*

*Geophys. Res. Lett., 42, 8702– 8710, doi:10.1002/2015GL065846.*

*EMPA-2013 for HFOs: Vollmer et al., Environ. Sci. Technol. 2015, 49, 5, 2703–2708.*

*SIO-05, SIO-14: Prinn et al., J. Geophys. Res., 105, 17,751-17,792, 2000, and Prinn et al, Earth Syst. Sci. Data, 10, 985–1018, 2018.*

The references were added to all scales in table 1.

**Table 1.** System precision ($1\sigma$) of the investigated substances treated as a training set of the TOF-MS used for the weekly whole air sampling (prc (TOF_Lab)) and of the TOF-MS used for the in situ measurements (prc (TOF_in situ)) and their calibration scales.

| Compound | Scale | prc (TOF_Lab) | prc (TOF_in situ) |
|---|---|---|---|
| HFC-32 | SIO-07[a] | 8.2 % | 2 % |
| HFC-125 | SIO-14[a] | 1.4 % | 0.9 % |
| HFC-143a | SIO-07[a] | 0.9 % | 1.7 % |
| PFC-318 | SIO-14[a] | 0.7 % | 3.3 % |
| HFC-152a | SIO-05[a] | 0.9 % | 1 % |
| HFC-227ea | SIO-14[a] | 7.1 % | - |
| HCFC-142b | SIO-05[a] | 0.3 % | 0.5 % |
| HCFC-133a | EMPA-13[b] | 2.8 % | 3.2 % |
| HFC-245fa | SIO-14[a] | 1.6 % | 4.3 % |
| HCFC-141b | SIO-05[a] | 0.8 % | 0.5 % |
| CFC-113 | SIO-05[a] | 4.4 % | 0.4 % |
| HFO-1234yf | METAS-17[a,c] | 18.2 % | 14 % |
| HFO-1234ze(E) | EMPA-13[d] | 6.9 % | 15.6 % |
| HCFO-1233zd(E) | EMPA-13[d] | 7.9 % | 14.1 % |

[a](Prinn et al. (2000), Prinn et al. (2018)), [b](Guillevic et al., 2018), [c](Vollmer et al., 2015b), [d](Vollmer et al., 2015a)

*25. l. 195: you probably need "the" in front of all "MAPE", check through the manuscript. I would add the equation for the computation or a reference (e.g. the Wiki page).*

Changed that and added article where necessary.

We added the equation for the calculation of the MAPE as suggested.

235  the mean absolute percentage error (MAPE), defined as:

$$\frac{1}{n} \cdot \sum_{t=1}^{n} \left| \frac{O(t) - F(t)}{O(t)} \cdot 100 \right| \qquad (6)$$

where n is the number of observations, F(t) is the predicted data as orthogonal distance regression fit forced through the origin and  O(t) is the data of the observed peak areas. The $r^2$ and the MAPE

**26.** *l. 199: "Except for HFC-227ea"*

Done.

**27.** *Section 3.2.1, general question: could you find explanations for the outlier rRFevalu data points?*

One possible explanation is the influence of water, which is the case for HCFC-141b using data of the whole air flask sampling. We commented on that in the revised version of manuscript. Other explanations do need further investigations.

> 210  trifluoroethane, $CCl_2FCClF_2$, CAS 76-13-1) in the case of the laboratory system. In these cases, water influences the signal intensity of the two compounds in the analysis in the laboratory system. Comparing them to their own intensities, as it is used in the direct calibrated analysis, they still show the mentioned precision. Due to the indirect calibration method, this change in signal intensity leads to an incomparability with other compounds not influenced by water vapor.

**28.** *l. 200-201, I would try to reformulate in an easier way. E.g.: "To test which pairs of substances produce the highest correlations, all possible pairs of substances have been tested. The obtained values for r2 and MAPE are shown in Fig 3".*

Changed as suggested.

**29.** *l. 208, typo: "all cases where HFC-152a is involved."*

Done.

**30.** *Also, it seems to me more to be a drift in the rRF value, that started before the change in standard tank, and stabilised after some runs of the new standard. Such a drift (albeit much smaller) can also be seen in the HFC-125 data points. So I'm really not sure that you can link this for the standard tank change. I would remove this sentence.*

The reviewer is correct. The change in rRF occurs before the change of standard. We have no explanation, why these drifts occur. We have made this clearer by changing the wording as follows:

> change of standard as a dashed vertical line. While for most combinations , the $rRF$ determined for the different standards do not differ significantly, a large discrepancy is found in all cases here HFC-152a is involved does not show a systematic change, the $rRF$ of HCFC-133a relative to HFC-152a shows a significant shift. However, this shift in $rRF$ started before the change
> 255  of standard and is thus obviously not related to an inconsistent calibration in the two standard gases used. The reason for this change shift is not known ., but this is illustrative of the limitations of the indirect calibration method. Under such extreme cases, strong shifts would be observed in the atmospheric measurements and such shifts should thus be treated with care. For

**31.** *l. 211: maybe you can comment on why HFC-227ea and HFC-245fa? HFC-227ea seems logical to be a bad one, as its measurement standard deviation given in Table 1 is one the highest. However why HFC-245fa? Or, alternatively, you can explain later why some are good ones?*

We chose to discuss HFC-227ea and HFC-245fa as substances less suited for the approach to demonstrate what the possible effects are. We agree that the poorer performance of HFC-227ea seemed likely given the poorer measurement precision, while this was somewhat unexpected in the case of HFC-245fa. However, based on the training substances included, we currently cannot explain why some pairs of substances are "good ones". The example thus shows that measurement precision is not a sufficient selection criterion which is the reason why we performed the additional statistical analyses.

**32.** *l. 220-222: "To quantify the differences between the selection of data of main reference and test substance via main reference substance and an evaluation substance we compared the relative standard deviations of the resulting filtered data sets." I don't understand this sentence. Please clarify. You may also want to cut into smaller sentences. Maybe, adding the equation you use will help to understand what you compute here.*

*Usually there are two quantitative values to characterise a result: its standard deviation, which reflect the random noise, and the average difference between two values (usually a test value and a reference value), which is a systematic bias. A bias not equal to zero means that the method causes a systematic error.*

*Now based on Fig 5, maybe what you want to express here is a precision loss, that you express via the difference in standard deviation? If this is really the case, here is my suggestion:*

*"To quantify the precision loss between direct calibration and calibration via a transfer substance, we compare the relative standard deviations of the resulting filtered data sets.", or something similar.*

The reviewer is right that we want to express is the loss in precision and we reworded the statement as suggested.

We rephrased the sentence as follows:

> and test substance. To quantify the  precision loss between direct calibration and calibration via an evaluation substance, we compared
> 275   the relative standard deviations of the resulting dataset as follows: (i) the $rRF_{test}$-dataset, applying the 10 % filter criterion directly (Fig. 5 (c) and (g)), and (ii) the $rRF_{test}$-dataset, using the data points which are selected via the residual $rRF_{evalu}$-data points applying the 10 % filter criterion (Fig. 5 (b) and (f)). This is shown for all substance combinations in

**33.** *Another important quantity to evaluate is if your method creates a bias or not? i.e. what is the average value of the distance (or difference) between the true and reconstructed value? It should be (close to) zero to show no bias. (cf see below comment on Table 3)*

See point 37.

**34.** *l. 237: if you mean precision loss, use: "the difference between the standard deviations".*

Done.

**35.** *l. 241: "As test cases to apply the indirect calibration method, we chose..." or "As test cases to be applied the indirect calibration method, ...".*

Done.

**36.** *l. 243: "mole fractions of HFC-227ea show..."*

Done.

**37.** *Table 3: average relative difference: this is your metric for the bias, right? Please write the equation somewhere in the text (e.g. around l. 245). Also: usually, if the bias or systematic offset value is within the 2 sigma standard deviation, it means within uncertainty, there is no bias. This is an important point to show here. But in Table 3, the "av. rel. difference" value is systematically more than the value of "standard deviation". Can you comment on this?*

Addition: We noticed an error. Due to new calculations, we decide to choose here the MAPE, as well. Even it is the calculation of the average relative difference we mentioned here before ( sum( $|(X_{direkt} - X_{indirect})/X_{direct}|$ ) * 100 ). Regarding to equation 6, O(t) is the direct calculated data and F(t) is the indirect calculated data.. The standard deviations shall explain, how much the relative deviation of each data point spread. So it is possible that they exhibit a larger amount as the MAPE (if the relative deviations show a high variability), or they show a small amount if it is a constant systematically error. Also the standard deviation can increase over too large chosen time periods, where the rRF exhibits maybe long term drifts.

**38.** *l. 275: typo: "HFO-1234yf"*

Done.

**39.** *l. 276: concordance of tenses, "increased continuously up to 100%"*

Done.

**40.** l. 311-312: "Further, it is likely that using reference species with similar retention times as the target species provides more stable results." Can you give an example here? No retention time data are provided.

While we tried to select reference species with similar retention times as the target species, we did not find any evidence that the quality of the indirect calibration is affected by the differences in retention times. We added a comment on that:

> use an evaluation substance to select periods when relative responses of the measurement system are rather stable. Further, it is likely that using reference species with similar retention times as the target species provides more stable results, which should be investigated with a larger number of training substances. The training dataset used in this work could not confirm
>
> 385 that. By analysing correlations and variabilities of the relative responses, we identify the combination of a main reference and an evaluation substance which  yield the minimum number of rejected data points of different target gases. Furthermore, we have chosen to include only time periods where the relative response of the reference substance

*41.* l. 313: "good results" is subjective. Maybe use a quantitative value instead, e.g. "which yield the minimum number of rejected data points".

Changed as suggested.

*42.* l. 330, typo: "is the measurement", "which are expected".

Done.

*43.* Your data show a rRF that is mostly not stable over time. Can you discuss the possibility to use a running-mean rRF value over time, instead of assuming a constant value over a short time period? Also, at least for some time periods, could you assign a (hardware?) cause to the non-stable rRF?

Using a running mean rRF over time is no option, due to systematical changes or drifts on the system. This would allow no qualification. We assume that non-stable rRFs are mostly caused by hardware issues. So this would be a good initiation to compare the method on other systems, which do maybe preserve other hardware constitutions.

Hardware issues can be assumed, when receiving a large standard deviation for the single relative deviations. A small standard deviation will mean that the system shows a systematically error.

*44.* Figure 5, legend: "Illustration of data selection for the weekly flask sampling measurements..."

Done.

---

## Author Comment (AC4)

**Reply to RC 4**

**Manuscript information:**

- Title: Non-target analysis using a gas chromatography with time-of-flight mass spectrometry: application to time series of fourth generation synthetic halocarbons at Taunus Observatory (Germany).
- Authors: Fides Lefrancois, Markus Jesswein, Markus Thoma, Andreas Engel, Kieran Stanley, and Tanja Schuck
- MS No.: amt-2020-488
- MS type: Research article
- Iteration: Final response (AMT Discussions)

We would like to thank Anja Claude for the constructive and detailed comments.

**Point-by-Point reply:**

1. *Close work with European metric institutions has brought up a discussion on "correct vocabulary". Among this the terminus "mole fraction". As "mole" is a unit, NMIs requested the use of "amount fraction" instead und the correct unit would be "nmol/mol" (instead of ppt) - just a remark, as I came across this discussion often recently.*

We agree that the correct unit would be pmol/mol. As the standards we use are prepared gravimetrically, we think that the wording "mole fraction" describes the measured quantities more precise than "amount fraction". For simplicity we will use ppt rather than pmol/mol and have added an explanatory sentence to the manuscript:

> ibration standard. Due to the low mole fractions of the investigated substances (range of picomole per mole; pmol mol$^{-1}$,
> 95  or hereafter, parts per trillion; ppt), a cryofocussing sample loop unit is used to

2. *Introduction: l. 47, "These hydro(chloro-)fluoroolefines are the so-called fourth generation of synthetic halocarbons…" - there are no other fourth generation synthetic halocarbons?*

We changed the sentence as follows:

> 50  HFC-1234ze(E), CAS 29118-24-9), and HCFO-1233zd(E) (E-1-chloro-3,3,3-trifluoroprop-1-ene, trans-CF$_3$(
> HCFC-1233zd(E), CAS 102687-65-0).  In the following we will use the H(C)FO nomenclature for the hydro(chloro-
> )fluoroolefines, as the HFC-nomenclature is not made for compounds with a double bond. These H(C)FOs are examples of
> the so-called fourth generation of synthetic halocarbons. Due to their carbon double bond,  HFOs and HCFOs

**3.** *l.51: "However, some HFO, as some HFC and HCFC, can form the very persistent and toxic trifluoroacetic acid (TFA) as…" check commas and the use of "some " Are the three substances of this paper among the TFA forming HFOs?*

We clarified this as follows:

> have an ODP of zero (Patten and Wuebbles, 2010; Orkin et al., 2014). However, some HFOs, e.g. the HFO-1234yf, and also some HFCs and HCFCs, can form the very persistent  trifluoroacetic acid (TFA), as the main breakdown product in the atmosphere (Burkholder et al., 2015). TFA
> 60  can accumulate in water and soil and can become moderately toxic to organisms (Ellis et al., 2001;

**4.** *l. 63 & 64: I think it has to be "Section" with "S" à please check!*

We will leave this typesetting question to the editorial team and pay close attention during final proofreading.

**5.** *l.70: "locations of industry" à maybe better "industrial areas" ?*

Changed as suggested.

**6.** *Section 2: l. 77: What does "approximately weekly" mean?*

This means that this sampling is not strictly weekly but we try to collect samples at different times of day and on different weekdays. In addition, sampling is done manually and is coordinated with other maintenance tasks at the site.

**7.** *l. 86: " …the sample loop is heated to approx. 200 °C…" - approximately?*

This has been changed in the manuscript to not use the contraction (approx.) but approximately instead.

**8.** *l. 99 "mud dauber" …an insect screen, I guess?*

We added that the mud dauber is also used as insect screen.

**9.** *l. 107ff: Are the sample inlets mounted in a separate heated box or are all parts in one single box? The description for the continuous instrument jumps a bit. Starting with the sampling procedure (flows, sample volume, desorption temperature) you continue with the unit set up (heated box, materials) and then return to the flushing procedure. The adsorption temperature is the same as in 2.2. (-80°C)? Streamlining this paragraph might improve the reading.*

We improved this section regarding the reviewer's suggestion as follows:
* * *
110  **2.3  Continuous in situ measurements**

Continuous in situ measurements, with ambient air sampling every two hours, were started at TOB in May 2018. The air intake is a 3/8" stainless steel inlet line, located 12 m above the ground, mounted outside a laboratory container with a goose-neck inlet. It uses a downstream diaphragm pump to continuously pull air from the inlet into the laboratory. , where the measurement system is located. This air intake line is heated to 70 °C to avoid condensation and

115  freezing. This is done by heating cables installed at the inlet line and extra insulation surrounding the whole line. To prevent the intrusion of particles, a mud dauber (Swagelok SS-MD-4), also used as insect screen, is installed at the open end of the inlet line. In the laboratory, the inlet line is connected to one of five sample inlets, mounted at a heated box (80 °C), connected via 1/8" quick connectors (Swagelok). Inside the heated box, sample inlets are connected to a 10-position-selector (model EUTA-2SD10MWE, Vici Valco Inc., USA) with 1/8" stainless steel tubing. A $Mg(ClO_4)_2$ dryer (similar to the Goethe

120  University laboratory setup; section 2.2) and a 4-port 2-position valve (model D4UWE, Vici Valco Inc., USA) are mounted inside the heated box and are connected via 1/8" stainless steel tubing. Directly before each measurement, the dryer and tubing of the system are purged and conditioned for 1 min at a flow of 100 mL min$^{-1}$ using the subsequent sample (air, calibration gas, etc.), bypassing the sample loop. Details are described by Obersteiner et al. (2016a) and are only briefly reviewed here. Halogenated trace gases are analysed using a  GC (Agilent 7890B), a TOF-MS (model EI-003, Tofwerk

125  AG, Switzerland), and a preceding enrichment unit, which is similar to the enrichment unit used in the laboratory. , using -80 °C for adsorption temperature, as well. For each measurement, approximately 500 mL of air are enriched in the sample loop at a sample flow of 80 mL  min$^{-1}$. To determine the exact volume of enriched air, a mass flow controller (MFC; EL-FLOW F-201CM, Bronkhorst) and a pressure sensor (Baratron 626, 0-1000 mbar, accuracy including non-linearity 0.25 % of reading, MKS Instruments, Germany) are used.

130  The sample loop is flash heated to about 220 °C for 120 seconds during sample desorption. Purified helium is used as carrier gas (quality 6.0, purification System: Vici Valco HP2). ~~Samples are dried using a trap to remove water vapour. Sample inlets are mounted inside a heated box (80 C) and are connected via 1/8" quick connectors (Swagelok). The dryer, a 10-position-selector (model EUTA-2SD10MWE, Vici Valco Inc., USA), and a 4-port 2-position valve (model D4UWE, Vici Valco Inc., USA) are mounted inside the heated box and are connected via 1/8" stainless steel tubing. Directly before each measurement, the dryer~~

135
* * *
**10.** *l. 133 : You neglect data for calibration intervals which deviate more than the weekly 1s-precision, ok – I am just interested: Did you ever take in account/discuss to add the additional error to the uncertainty?*

Yes, we have considered this. However, we decided against including such data points with poorer precision as this would e.g. be problematic when separating the data set into baseline data and outliers caused by local pollution. As this is not at the focus of this paper, we do not add a discussion on this.

*11.     Section 3: Figure 1: as I understand this figure is explanatory only, in order to describe the method how stable periods were defined. Therefore the plotted compounds are not mentioned. Nevertheless, the question bothered me while looking at the plot, what substances are plotted. Later on, in Figures 5 and 7 you show the similar plots again. I wonder, if you could combine those?*

We added comments on that to clarify this. We agree, that Figures 5 and 7 show the same information for real data as is shown in Figure 1 for an artificial dataset. However, we think that for the reader it is more understandable to explain the procedure and the idea step by step and would therefore prefer to leave the figures as they are.

> **Figure 1.** Schematic example using a random data set of the identification of periods with constant $rRF$ for an undetectable substance in the calibration standard. Panels (a) and (b) show the calculated $rRF_{evalu}$ of a known main reference and a known evaluation substance. Panel

> 215  change in signal intensity leads to an incomparability with other compounds not by water influenced.
>      Fig. 1 shows a schematic example for the identification of periods of stable $rRF$, where a random data-set was created.
>      Panel (a) shows the $rRF_{evalu}$ for two arbitrary substances, a so called main reference and an evaluation substance, which

*12.     l. 181: Why did you choose the stability criterion to be 10%?*

The value of 10% is somewhat arbitrary and was chosen as a compromise to have a sufficient number of measurement points while at the same time exclude data with a large measurement uncertainty.

We modified the manuscript as follows:

> 220  as grey data points in panel (b)). If more than one measurement has the same number of matching data points, the case with the lowest standard deviation is selected. In our application, we have arbitrarily chosen a maximum deviation of 10 % as a selection criterion, as it allowed to retain a sufficient number of measurements while still eliminating data which would have
>      particularly large uncertainty. Depending on the stability of the instrument and the desired results, different criteria could be chosen. Allowing for larger deviations would result in retaining more data with larger uncertainties, while applying a more
> 225  stringent criterion would result in a dataset with less data yet likely also lower uncertainties. In panels (c) and (d) the evaluation

**13.** *l. 195 ff. I had to read this sentence several time. Just to be sure: MAPE is the difference "A_measured – A_odr", with "A_odr" being the ODR fit of peak areas forced through the origin*

We clarified this as follows by adding also the formula used to calculate MAPE:

235   the mean absolute percentage error (MAPE), defined as:

$$\frac{1}{n} \cdot \sum_{t=1}^{n} \left| \frac{O(t) - F(t)}{O(t)} \cdot 100 \right| \tag{6}$$

where n is the number of observations, F(t) is the predicted data as orthogonal distance regression fit forced through the origin and  O(t) is the data of the observed peak areas. The $r^2$ and the MAPE

**14.** *l. 210: "Using HFC-227ea and…." You already present a single result of the analysis that follows in the next paragraph. I found this is confusing. From my point of view you can delete this sentence. Figure 5 (&7): I understood that you derived stable periods from calibration measurements as well. However, in the caption of Figure 5 you mention "data of the weekly flask sampling measurements". Please clarify.*

We changed l. 210 as follows, to make the section more clear. Using this sentence, we want to point out the importance of the selection of appropriate substances to receive highest possible amount of data.

the number of measurements after selecting the periods of constant $rRF$ should remain as high as possible.  E.g. using HFC-227ea and HFC-245fa as evaluation substances in combination with HFC-143a as main reference substance or using HFC-245fa as a main reference, more than half of the calibration measurements are excluded, as shown in Fig. 5, panels (d) and (h). As this leads to a significant decrease in the number of air measurements for which an indirect calibration value can 265   be derived, these substances are also less suitable as reference substances.

"Data of weekly flask sampling measurements" means that rRF is calculated using the calibration measurements, used during the flask sampling measurements. We clarified this as follows:

**Figure 5.** Illustration of  constant $rRF$ data selection, using the standard calibration measurements during the weekly flask sampling measurements, using two different main reference  substances (HFC-143a (a-d), HFC-245fa (e-h)) and an

**15.** *l. 221ff: "To quantify the differences…" I do not understand this sentence, please explain. As a consequence please also explain panels d and h in Figures 5 and 7. Not clear to me.*

We clarified this as follows:

and test substance. To quantify the  precision loss between direct calibration and calibration via an evaluation substance, we compared
275   the relative standard deviations of the resulting  dataset as follows: (i) the $rRF_{test}$-dataset, applying the 10 % filter criterion directly (Fig. 5 (c) and (g)), and (ii) the $rRF_{test}$-dataset, using the data points which are selected via the residual $rRF_{evalu}$-data points applying the 10 % filter criterion (Fig. 5 (b) and (f)). This is shown for all substance combinations in

**16.** *l. 229ff: "In summary…" These are the characteristics you expect from your relative Response factor to achieve a good calibration and which set the frame for your checks you presented in the preceding lines. Might be good to have it earlier in the text?*

In the preceding paragraphs of the subsection we have discussed several aspects of the variability of the rRF. We therefore think that it makes sense at this point to briefly summarize the characteristics that are needed right before discussing our choice of substances. At this point we summarize the characteristics discussed in the previous paragraphs of this subsection.

**17.** *l. 237: "Using HFC-125 as evaluation substance with HFC-143a, the difference standard deviations of the mean rRF selected via the test substances and selected via itself ranges between 1 and 10 %." Do you mean "different"?*

Yes, we meant different and this has been altered in the revised manuscript.

**18.** *l. 243: "In this test case, mole fractions of HFC-227ea shows the best correlation …" "show" instead of "shows".*

Altered to "show".

**19.** *l. 249: You end this paragraph with a statement about the importance of having a constant sensitivity. You take up this point but then do not really discuss it. It would be nice to have all the presented results of 3.2.2 "wrapped up" at this point in final short summery*

We think that this has been discussed in the previous paragraphs and refer to the summarizing sentence that is an introduction into the final parts of subsection 3.2.2 (cf. comment 16. above). We therefore prefer to end this subsection with the statement about the importance of a constant response factor.

**20.**   *Table 2: With standard deviation you mean the standard deviation of the average relative difference? Is it this really necessary to have this table or could you add this information into Fig.6. ?*

Standard deviation here indeed refers to the standard deviation of the relative deviations of the data points. An explanation has been added to the table caption.

Adding more information into Figure 6 would result in an even more crowded figure. We think that therefore a table is better suited and, in addition, it facilitates the comparison of the numbers for the three compounds.

When reviewing the data, we discovered an error in the numbers in Table 2 and this was corrected.

and HFC-245fa poorer results with $r^2 = 0.79$ and $r^2 = 0.63$, respectively, are obtained. The  MAPE, where F(t) is now defined as the indirect calculated mole fractions, and O(t) defined as direct calculated mole fractions (cf. eq. 6) are given in Table 2. Table 2 shows also the standard deviation of the relative deviations between indirect and calculated mole fractions, to show the spread of the differences between direct and indirect calculated mole fractions. The

305    $rRF$ of HFC-245fa as evaluation substance and HFC-143a as main reference substance has also less than 50 % of selected data within the 10 %-filter (cf. Fig. 5), which means that the calculation is applied to a large portion of data for which the criterion of a constant $rRF$ was not met. This underlines how crucial the assumption of constant instrumental sensitivity is for the indirect calibration method.

**Table 2.**  The mean percentage error (MAPE) of directly and indirectly calculated mole fractions and standard deviations of the relative deviations of each data point (direct vs. indirect) for the whole air flask sample GC-MS measurements for the period from October 2013 to December 2018 (cf. Fig. 6). As main reference HFC-143a, as evaluation reference HFC-125 is used.

| compound |  MAPE | standard deviation |
|---|---|---|
| HFC-32 | 7.5 % | 6.4 % |
| HFC-227ea | 11.1 % | 12 % |
| HFC-245fa | 8.8 % | 9.0 % |

21. *Figures 8 and 6: When you calculated the amount fractions via the indirect way you applied the method for data in the stable periods (as derived in Figure 5 and 7 respectively). The constant rRFs you used to "attach" the test substances to the reference substance are determined using the average rRF in the calibration measurements also during the stable times? This is not clear to me? Why did you use the measurement precision as error bars? How does this reflect the uncertainty factors of the different calibration methods?*

It is correct that average rRFs are only determined based on data from periods of stable rRF, as described in section 3.2 Method evaluation.

It is correct that in Figure 8 the error bars represent the standard deviation of the measurements of one day. The data points shown represent daily averages. Therefore, these error bars should not be taken as a measure of the absolute error but rather as an atmospheric variability. We specified this in the modified figure caption:

**Figure 8.** As Fig. 6, but for the continuous in situ measurements at Taunus Observatory using GC-MS and for the following compounds: HFC-125, HFC-143a, and HFC-245fa, calculated directly (yellow symbols) and indirectly ( blue symbols). Here, HFC-152a was used as main reference substance, PFC-318 as evaluation substance to select data with constant $rRF$. For reasons of simplified illustration daily means are shown. The error bars indicate the standard deviations of the measurements of one day and thus reflect the daily atmospheric variability rather and do not include systematic errors due to the indirect calibration method.

22. *Figure 6: What happens if you omit the two single data points in January 2018 for HFC-32? Any idea what happened here?*

Omitting the two high values in the case of HFC-32 in Figure 6 results in a value of r² closer to 1 and a slope closer to 1. Because we do not have any indications of a technical issue causing these outliers, we did not remove them. The correlations are used as a pre-selection criterion. As a large number of outliers worsens the correlation, the example of HFC-32 shows the necessity for such a criterion.

23. *Figure 8: What happens to the regression lines when you omit the high amount fractions?*

For data shown in Figure 8, omitting the high values leads to smaller values of r². In fact, this shows the limitations of the method as discussed in the conclusion section. The method is not well suited for the detection of small variabilities or of small trends due to its rather large uncertainty.

24. *Tables 2+3: Do you use this deviation to derive the uncertainty of the method?*

Yes. We use this deviation to quantify the uncertainty of this method. Because we cannot find other ways to compare the unknown data points with known data points, we use this approach (using the test datasets) to find an appropriate quantification for the determination of the uncertainty.

25.  *l. 262: "This is caused by long-term drifts…" What does this tell you about the possible errors arising from differences between the evaluation and test substances, even though you have filtered out periods with a larger variability?*

This confirms that the method is not applicable for small long-term trends, as we point out in the conclusions section.

26.  *Section 4: Figures 9 and 10: maybe you keep the zoomed in plot, only? What do the error bars represent?*

To visualize our whole data records, we prefer to leave both plots in the paper. We explained the error bars as follows:

> **Figure 9.** Time series of mole fractions of HFO-1234yf (a), HFO-1234ze(E) (b), and HCFO-1233zd(E) (c) at TOB. Symbol shape indicates flask sampling (diamonds) and in situ measurement (circles). Grey symbols represent mole fractions below detection limit (cf. Table 2 and 4). Data of weekly flask sampling are indirectly calibrated mole fractions before January 2018 and directly calibrated values after. For the in situ measurements, indirectly and directly calibrated mole fractions are indicated by color (orange and yellow). Note the change in x-axis scaling after January 2018 because of increased data frequency. Data of in situ measurements show daily means. Errorbars for weekly flask sampling show the 1 $\sigma$ standard deviation of the measurements for one sampling. Errorbars for the in situ measurements show the standard deviation of the average of measurements over one day.

27.  *l. 291: "These larger amounts could be…." So, this is the effect of non-linearity or a larger integration error for the small calibration peak?*

Up to now we have not found any indications of non-linearity for the investigated H(C)FOs but it seems likely that the large uncertainty results from the size of the peak. We have reworded this for more clarity:

> large deviation with maximal 44 % for the data in 2018. These larger  deviations could be caused by the
> 360   small peak of HFO-1234ze(E) in the calibration gas used from May 2018 to March 2019. The mean values of both substances,

28.  *Conclusion: You have presented methods to derive the best possible reference substances for your indirect calibration and you evaluate this indirect calibration procedure regarding its performance. It would have been nice to see an assessment of how uncertainties arising from the determination of the rRF and evaluation of stable periods (by the evaluation) are reflected in the results of the test compound analysis. E.g. do expected errors match with observed differences between indirect and direct calibration?*

In this work we focused on the development of this approach and on demonstrating its application to atmospheric measurements. To disentangle the various sources of uncertainties, further data evaluation would be necessary, for example applying the method to a larger number of substances. In addition, we still need a better understanding of the causes of the periods of unstable rRF.